# Fine-tuning of auxin homeostasis governs the transition from floral stem cell maintenance to gynoecium formation

Nobutoshi Yamaguchi[1,2], Jiangbo Huang[3,4], Yifeng Xu[3], Keitaro Tanoi [2,5] & Toshiro Ito[1,3,4]

To ensure successful plant reproduction and crop production, the spatial and temporal control of the termination of the floral meristem must be coordinated. In *Arabidopsis*, the timing of this termination is determined by AGAMOUS (AG). Following its termination, the floral meristem underdoes gynoecium formation. A direct target of AG, *CRABS CLAW* (*CRC*), is involved in both floral meristem determinacy and gynoecium development. However, how floral meristem termination is coordinated with gynoecium formation is not understood. Here, we identify a mechanistic link between floral meristem termination and gynoecium development through fine-tuning of auxin homeostasis by CRC. CRC controls auxin homeostasis in the medial region of the developing gynoecium to generate proper auxin maxima. This regulation partially occurs via direct transcriptional repression of *TORNADO2* (*TRN2*) by CRC. Plasma membrane-localized TRN2 modulates auxin homeostasis. We propose a model describing how regulation of auxin homeostasis mediates the transition from floral meristem termination to gynoecium development.

[1] Biological Sciences, Nara Institute of Science and Technology, 8916-5 Takayama, Ikoma, Nara 630-0192, Japan. [2] Precursory Research for Embryonic Science and Technology, Japan Science and Technology Agency, 4-1-8 Honcho, Kawaguchi-shi, Saitama 332-0012, Japan. [3] Temasek Life Sciences Laboratory, 1 Research Link, National University of Singapore, Singapore 117604, Singapore. [4] Department of Biological Sciences, National University of Singapore, Singapore 117543, Singapore. [5] Graduate School of Agricultural and Life Sciences, The University of Tokyo, 1-1-1 Yayoi, Bunkyo-ku, Tokyo 113-8657, Japan. Nobutoshi Yamaguchi and Jiangbo Huang contributed equally to this work. Correspondence and requests for materials should be addressed to T.I. (email: itot@bs.naist.jp)

The inflorescence meristem undergoes two balanced, antagonistic processes: self-renewal of stem cells to maintain their proliferation activity and recruitment of peripheral cells for organogenesis to form flowers[1–3]. These two mechanisms occur continuously during the reproductive phase. Unlike indeterminate vegetative and inflorescence meristems, floral meristems exhibit a dynamic balance between cell proliferation and organ initiation. Floral meristems sequentially form four types of floral organs without undergoing elongation of the internode between floral organs. Floral meristems are genetically programmed to be terminated at a specific developmental stage after floral organ formation[4,5]. Thus, floral meristems are determinate. This precise termination process allows floral meristems to form a female reproductive structure referred to as a gynoecium[6,7].

**Fig. 1** Indeterminate phenotype of the *crc knu* double mutant is caused by WUS activity. **a–d** The floral meristem activity marker *WUS* mRNA in longitudinal sections of wild-type (**a**) *crc-1* (**b**) *knu-1* (**c**), and *crc-1 knu-1* (**d**) floral buds at developmental stage 6. **e–h** Flower phenotypes in *wus-1* (**e**), *wus-1 crc-1* (**f**), *wus-1 knu-1* (**g**), and *wus-1 crc-1 knu-1* (**h**). **i, j** Morphology of *crc-1 knu-1* (**i**) and *wus-1 −/+;crc-1 knu-1* (**j**) fruits. Top: close-up views of fruit tips. Bottom: shapes of whole fruits. Arrowheads indicate stigma structures. Two stigma-like structures were observed in the *wus-1 −/+ crc-1 knu-1* mutant due to carpel separation. Images are shown at the same magnification. **k** Quantification of mutant phenotype. *p*-values were calculated by $\chi^2$ test (*n* = 30). **l, m** Longitudinal sections of *crc-1 knu-1* (**l**) and *wus-1 −/+ crc-1 knu-1* (**m**) fruits. Asterisks indicate ectopic carpels. **n, o** Scanning electron micrographs of fruits with reiterations of floral organs in *crc-1 knu-1* (**n**) and *wus-1 −/+ crc-1 knu-1* (**o**) fruits. Bars = 50 μm in **a–d**; 5 mm in **e–h**; 1 cm in **i, j** bottom; 500 μm in **l, m**; and 1 mm in **n, o**

The transcriptional regulatory cascade underlying the termination of floral meristems has been characterized in the model plant *Arabidopsis thaliana*. Floral meristem termination must be completed during stage 6 of flower development when differential gynoecium growth along the medial-lateral and adaxial-abaxial axis has led to central invagination[8]. The C class gene *AGAMOUS*

(*AG*) plays a central role in this termination process, as well as the establishment of reproductive organ identity[9–12]. AG turns off the stem cell maintenance program by repressing the expression of the key meristem maintenance gene *WUSCHEL* (*WUS*) during stage 6 of flower development in a spatio-temporal-specific manner. *WUS* expression begins to decline when *AG* is

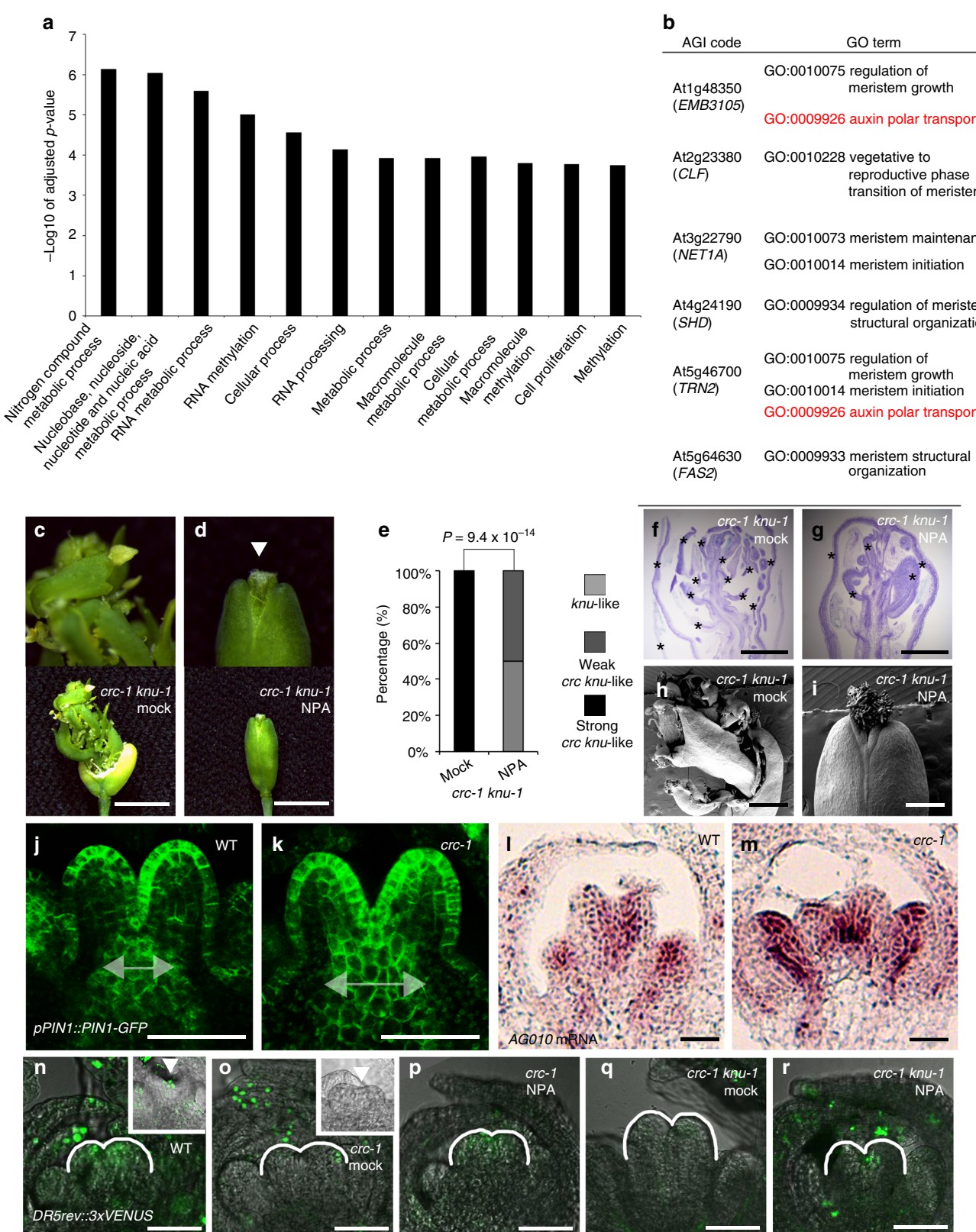

activated[13,14]. In *ag* mutants, *WUS* expression remains active beyond stage 6, which is sufficient to induce floral meristem indeterminacy[15,16]. Temporal *WUS* repression by AG is controlled by two different pathways. AG is necessary to recruit Polycomb Group protein (PcG) to methylate histone H3K27 at the *WUS* locus[13]. In addition, *WUS* repression by AG occurs at least partially through the C2H2 zinc-finger protein KNUCKLES (KNU)[17]. In *knu* mutants, obvious *WUS* expression continues after stage 6 in the developing gynoecium[18]. AG directly activates *KNU*, a process that requires eviction of the PcG complex, leading to cell cycle-dependent *KNU* induction[18,19]. This precise control of *KNU* activation helps determine the timing of floral meristem termination.

After termination of floral meristems, cells located at the center of stage 6 flowers become gynoecium, which consists of two fused carpels[6,7]. Previous studies demonstrated that auxin homeostasis contributes to the generation of developmentally important auxin maxima. Not only the well-known PIN-FORMED family of auxin carriers, but also other membrane proteins affect auxin maxima[20,21]. For example, *TORNADO2* (*TRN2*) encodes a transmembrane protein of the tetraspanin family, the loss-of-function mutant of which shows aberrant auxin maxima in leaves[22–25]. In flowers, *TRN2* is expressed in stigma at flowers[22,24,25]. Despite the critical importance of the local action of auxin in gynoecium development, its possible function when floral meristem is terminated is not known.

In this study, we obtained insights into a KNU-independent floral meristem termination pathway downstream of AG, which is coordinated by an AG direct target, *CRABS CLAW* (*CRC*)[26–31]. Previous findings reported that *crc knu* double mutant shows strong indeterminacy in the floral meristem[31] and *crc* mutant phenotype is partially rescued by an auxin transport inhibitor[32]. However, a mechanistic link between floral meristem termination and auxin-mediated gynoecium formation is not known. Here, we found that two direct AG target genes, *KNU* and *CRC*, synergistically regulate *WUS* repression. We further show that CRC represses the expression of *TRN2* and controls floral meristem determinacy. Our results highlight the molecular framework underlying the AG-mediated termination of floral meristems through auxin, which is required for subsequent gynoecium development.

## Results

**Indeterminacy in *crc knu* is caused by *WUS* misexpression.** Although *CRC* is a direct target of AG at stage 6 of flower development[11,27] (Supplementary Fig. 1), the *crc* single mutant has only a subtle or no indeterminate floral meristem phenotype[26]. Double mutants in genetic backgrounds with reduced *AG* expression or with mutations in the downstream target genes of AG show enhanced indeterminacy[29–31]. Because the *knu* single mutant exhibits an indeterminate phenotype[17], this

mutant provides a sensitized background for studying the role of CRC in floral meristem termination (Supplementary Fig. 2). Before conducting a detailed molecular analysis, we confirmed the previously reported aberrant phenotypes of the *crc knu* double mutant and parental lines[31] (Supplementary Fig. 2a–e). Compared to the wild type, style tissue is reduced in the *crc-1* mutant, leading to the presence of two unfused carpels at the apical region and two separate stigmas ($p = 9.5 \times 10^{-15}$) (Supplementary Fig. 2a, b, e). Wild-type and *crc-1* mutant fruits contain developing seeds enclosed by carpels (Supplementary Fig. 2a, b). By contrast, the *knu* mutants have bulged gynoecia surrounding additional whorls of floral organs (Supplementary Fig. 2c, f). *crc-1 knu-1* double mutant flowers exhibit synergistic indeterminate phenotypes ($p = 9.4 \times 10^{-14}$) (Supplementary Fig. 2d, e, g). Although a few fruits showed a weak phenotype without internode elongation, as shown in Supplementary Fig. 2h (categorized as weak *crc knu*), most *crc knu* fruits showed overgrowth of the indeterminate shoot phenotype (categorized as strong *crc knu*) (Supplementary Fig. 2d, e, g, i). This ectopic floral internode axis continued to elongate, with approximately ten rounds of carpel production (Supplementary Fig. 3a) and stamens occasionally appearing in a spiral pattern (Supplementary Fig. 2i, inset). Due to these phenotypic variations, we divided the mutant phenotypes into five categories based on their phenotypic severity and quantified the phenotypes in this and subsequent analyses (Supplementary Fig. 2e).

To address the role of *CRC* in floral meristem activity, we monitored the expression of *WUS* (Fig. 1a–d and Supplementary Fig. 3b–e). In stage 6 flowers, *WUS* expression was terminated in wild-type and *crc-1* plants (Fig. 1a, b). In the *knu-1* single mutant, weak but distinct *WUS* signals were detected in the centers of stage 6 and later flowers[18] (Fig. 1a–c and Supplementary Fig. 3b–d). A strong increase in *WUS* messenger RNA (mRNA) levels was detected in the *crc-1 knu-1* (Fig. 1a–d and Supplementary Fig. 3c–e).

To investigate whether the observed phenotype depends on the WUS activity, we next created *wus-1 crc-1 knu-1* triple mutants to remove WUS activity from the *crc knu* double mutant. The triple mutant flowers were indistinguishable from those of the *wus-1* single mutant (Fig. 1e–h). These results indicate that *wus* is epistatic to *crc* or *knu* in floral meristems. Furthermore, indeterminacy of the *wus-1 −/+ crc-1 knu-1* mutant was significantly rescued compared to the *crc-1 knu-1* double mutant ($p = 4.3 \times 10^{-8}$) (Fig. 1i–o), suggesting that the indeterminate phenotype of the *crc knu* mutant is caused by ectopic *WUS* expression.

**CRC controls auxin homeostasis and establishes auxin maxima.** Since synergistic effects are observed in *crc knu* double mutants, disruption of multiple pathways and drastic transcriptional changes are thought to occur in double mutants, but not in

**Fig. 2** Indeterminate phenotype of the *crc knu* double mutant is rescued by auxin transport inhibitor treatment. **a** GO terms enriched among the 210 upregulated gene in *crc knu*. For a list of these genes, see Supplementary Data 1 and 2. FDR cutoff of <0.03% was implemented. The −log10 adjusted *p*-values of all significant GO terms are shown. **b** List of the six genes with likely roles in metabolic process and meristem activity. Among the six genes, *EMB3105* and *TRN2* play a role in regulating 'auxin polar transport', as highlighted in red. **c, d** Morphology of mock-treated *crc-1 knu-1* (**c**) and NPA-treated *crc-1 knu-1* (**d**) fruits. Above: close-up views of fruit tips. Below: shapes of whole fruits. Arrowhead indicates stigma structures. **e** Quantification of mutant phenotype. *p*-values were calculated by $\chi^2$ test ($n = 30$). **f, g** Longitudinal section of mock-treated *crc-1 knu-1* (**f**) and NPA-treated *crc-1 knu-1* (**g**) fruits. Asterisks indicate carpels. **h, i** Scanning electron micrograph of *crc-1 knu-1* fruits. **h** Mock-treated *crc-1 knu-1* fruits with several reiterations of floral organs. **i** NPA-treated *crc-1 knu-1* fruits with reduced reiterations. **j, k** pPIN1::PIN1-GFP expression in wild-type (**j**) and *crc-1* (**k**) flowers. **l, m** In situ hybridization showing the distribution of the adaxial fate marker *ARGONAUTE10* (*AGO10*) in longitudinal sections of wild-type (**l**), and *crc-1* (**m**) floral buds at developmental stage 6. **n–r** DR5rev::3xVENUS expression in wild-type (**n**), mock-treated *crc-1* (**o**), NPA-treated *crc-1* (**p**), mock-treated *crc-1 knu-1* (**q**), and NPA-treated *crc-1 knu-1* (**r**) floral buds at developmental stage 6. Insets are closer inspection of floral buds. Arrowheads indicate medial region of gynoecium. Bars = 1 cm in **c, d** bottom; 500 μm in **f, g**; 1 mm in **h, i**; 50 μm in **j, k** and **n–o**; and 25 μm in **l, m**

single mutants. To explore the molecular basis for the CRC downstream pathway that affects *WUS* expression, we performed expression profiling of *knu-1* and *crc-1 knu-1*. This analysis identified 363 genes whose expression levels were altered more than 2-fold at a 99% confidence level in the *crc-1 knu-1* double mutant relative to the *knu-1* single mutant (Supplementary

Data 1). Among those genes, 210 and 153 genes were up- and downregulated in *crc knu*, respectively (Supplementary Fig. 4a and Supplementary Data 1).

To investigate the functions of these genes, we performed Gene Ontology (GO) term enrichment analysis using agriGO[33]. GO terms related to numerous metabolic processes were significantly

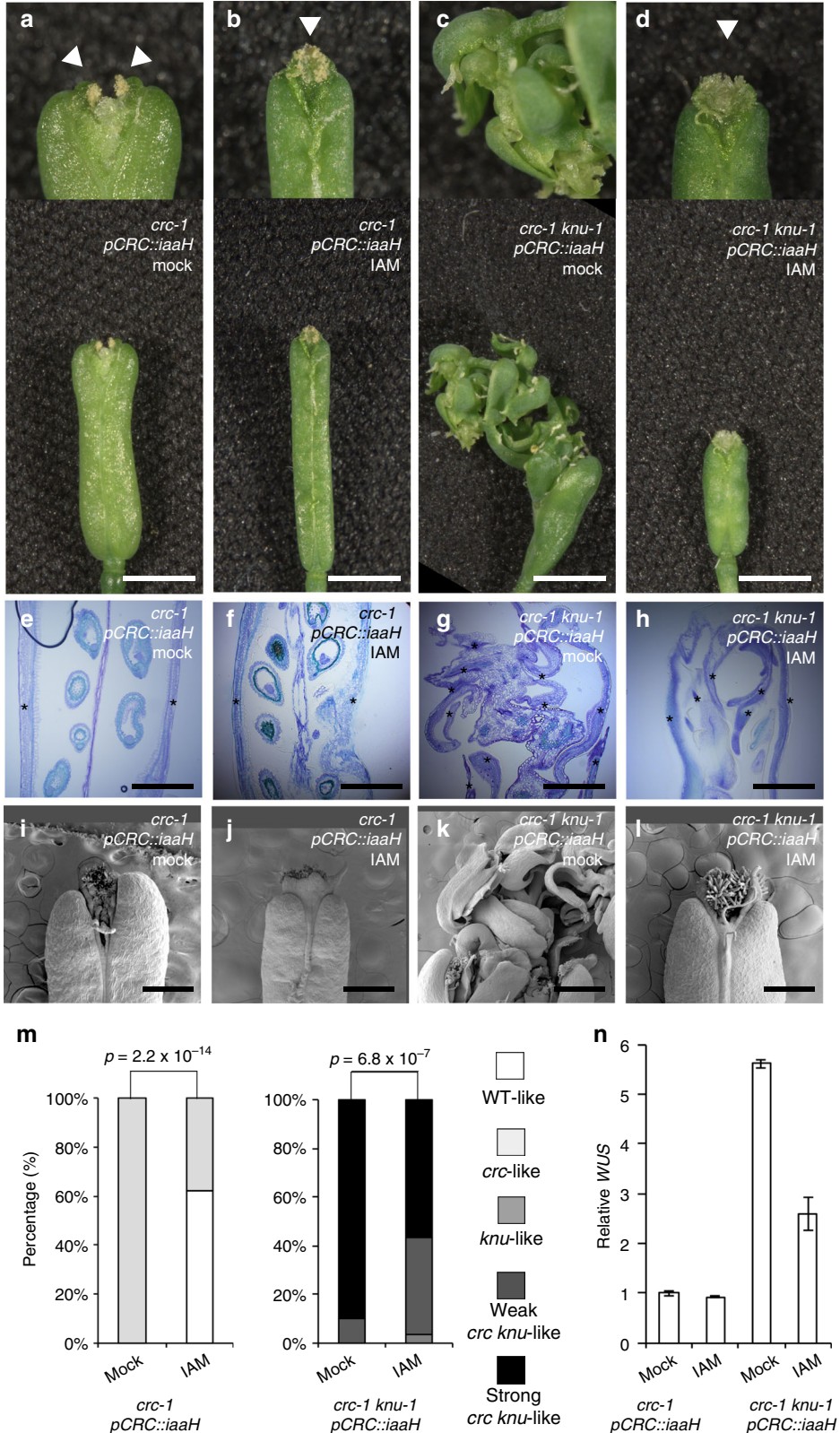

enriched in *crc knu* ($p < 0.0002$, FDR $<0.032$), and 123 up- or downregulated genes were involved in 'metabolic processes' (Fig. 2a, Supplementary Fig. 4b and Supplementary Data 1 and 2). To further determine which of the 123 metabolic genes regulate floral meristem determinacy, we determined which genes contained 'meristem' in their GO terms (Fig. 2b, Supplementary Data 2 and Table 1). Among these six genes, two encoded chromatin regulators, *CURLY LEAF* (*CLF*) and *FASCIATA2* (*FAS2*)[34,35] (Fig. 2b). A ribosomal protein (EMBRYO DEFECTIVE 3105 [EMB3105]), an actin-binding protein (NETWORKED 1A [NET1A]), a HEAT SHOCK PROTEIN 90-like protein (SHEPHERD [SHD]), and a membrane protein (TRN2) were also identified as possible CRC downstream targets[22,36–38] (Fig. 2b). Among these six genes, *EMB3105* and *TRN2* contained 'auxin polar transport' in their GO terms (Fig. 2b and Supplementary Table 1). This finding is not unexpected, since metabolic pathways are essential for producing secondary metabolites, including plant hormones[39]. Thus, we hypothesized that a previously hypothesized non-cell-autonomous signal[40] downstream of CRC may be auxin.

The style and stigma phenotypes observed in the *crc* mutant were reportedly rescued by the auxin transport inhibitor N-1-naphthylphthalamic acid (NPA) treatment[32] ($p = 3.4 \times 10^{-6}$), while wild-type and *knu* flowers were not significantly affected (Supplementary Figs. 5a–c and 6a–d). Similar effects were observed when we used 1-naphthoxyacetic acid (1-NOA), which inhibits auxin influx ($p = 3.1 \times 10^{-7}$) (Supplementary Fig. 5d–f). This rescue was not observed when we treated the *crc* mutant with the auxin signaling inhibitor *p*-chlorophenoxyisobutyric acid (PCIB) (Supplementary Fig. 5g–i). To further probe the biological importance of the CRC-mediated role of auxin transport in floral meristem determinacy, *crc-1 knu-1* plants were treated by inhibitors. NPA and 1-NOA treatments also brought the *crc knu* double mutant back to the *knu* mutant phenotype in terms of floral meristem determinacy (NPA: $p = 9.4 \times 10^{-14}$; 1-NOA: $p = 6.1 \times 10^{-13}$), while PCIB did not ($p = 0.31$) (Fig. 2c–i and Supplementary Fig. 7a–f). All of the inner structures in NPA-treated *crc-1 knu-1* plants were covered with two carpels, and the number of ectopic carpels was reduced, as observed in the *knu-1* mutant (Fig. 2f–i and Supplementary Fig. 6). These results suggest that CRC controls auxin transport, but not signaling required for floral meristem termination.

We next observed the green fluorescent protein (GFP)-tagged PIN1 line (*pPIN1::PIN1-GFP*[41–43]) to infer possible changes in auxin distribution. In wild-type floral buds at stage 6, PIN1-GFP signals appeared stronger on the medial domain of the gynoecium, which gives rise to the replum, style, and stigma (Fig. 2j). Basipetal PIN1 polarization caused auxin to move from the top to the bottom of the gynoecium at this stage[43] (Fig. 2k and Supplementary Fig. 8a, b). Consistent with the slight adaxialization of carpels observed in the *crc* mutant based on *ARGONAUTE10* (*AGO10*) and *PHABULOSA* (*PHB*) expression, PIN1 expression domain expanded laterally in the *crc* mutant, but no difference in its cellular localization was observed (Fig. 2j–m and Supplementary Fig. 8a–d). Furthermore, no obvious PIN1

expression and localization differences between the wild-type and *crc knu* mutants were observed at stage 5 flowers prior to floral meristem termination (Supplementary Fig. 8e, f). By stage 6, the PIN1 expression domain in *crc knu* was expanded as seen in the *crc* mutant (Supplementary Fig. 8g, h). Compared to the wild type, reduced expression of the auxin-response reporter *DR5rev::3xVENUS* was detected in the *crc* mutant at the medial domain of the gynoecium[43] (Fig. 2n, o). Since NPA treatment in *crc* and *crc knu* led to increased *DR5* signals, phenotypic changes observed in *crc* and *crc knu* treated with NPA are likely attributable to enhanced auxin maxima (Fig. 2o–r).

**Local auxin production rescues the indeterminacy in *crc knu*.** To investigate whether local overproduction of auxin is sufficient to bring *crc knu* back to *knu* mutant, we expressed the *Agrobacterium* indole acetamide (IAM) hydrolase (*iaaH*) gene under the control of the *CRC* promoter (*pCRC::iaaH*) in these mutants[44] (Fig. 3a–l). Since iaaH catalyzes the biosynthesis of bioactive auxin (indole-3-acetic acid) from IAM, *pCRC::iaaH* transgenic plants overproduced auxin in the peripheral region of the floral meristem only when we supplied them with the precursor IAM. In the absence of IAM, the *iaaH* transgene did not affect mutant phenotypes in either the *crc* or *crc knu* mutant background (Fig. 3a, c, e, g, i, k). Overproduction of auxin by IAM application in the peripheral regions of developing carpels triggered significant rescue of *crc* mutant phenotypes ($p = 2.2 \times 10^{-14}$) (Fig. 3m). IAM-treated *pCRC::iaaH* fruits exhibited fused carpels (Fig. 3a, b, e, f, i, j). Likewise, *pCRC::iaaH* activation also restored the phenotype of the *crc knu* double mutant to that of *knu* in terms of floral meristem determinacy ($p = 6.8 \times 10^{-7}$) (Fig. 3d, h, l, m). Unlike *crc knu* double mutants, IAM-treated *pCRC::iaaH crc knu* plants did not show overgrowth of indeterminate shoots (Fig. 3c, d, g, h, k, l). By contrast, we did not detect any phenotypic differences in *crc* or *crc knu* treated with the optimal concentration of auxin chemicals[45] (Supplementary Figs. 5j–r and 7g–o), suggesting that auxin maxima may be necessary for proper meristem determinacy. The strong *WUS* expression observed in *crc-1 knu-1* double mutants was attenuated when auxin was locally overproduced (Fig. 3n). Taken together with the NPA treatment result, our results uncover a possible link between local auxin homeostasis and floral meristem determinacy.

**CRC represses *TRN2* expression via a YABBY-binding site.** Based on our expression profiling analysis, we identified two auxin-related targets acting downstream of CRC. Since the role of *TRN2* in flower meristem determinacy is largely unknown and the biggest expression changes in *crc knu* were observed compared *knu* mutant among six candidate genes (Supplementary Fig. 9a–f), we focused on the *TRN2* gene as a putative direct target of CRC. We first investigated whether CRC regulates *TRN2* expression. *TRN2* was highly expressed throughout *crc-1* carpel primordia at stage 6 compared to wild type (Fig. 4a–c). Since the *TRN2*-misexpressing domain did not fully overlap with *CRC*

**Fig. 3** Rescue of *crc knu* indeterminacy by local overproduction of auxin. **a–d** Morphology of mock-treated *crc-1 pCRC::iaaH* (**a**), IAM-treated *crc-1 pCRC::iaaH* (**b**), mock-treated *crc-1 knu-1 pCRC::iaaH* (**c**), and IAM-treated *crc-1 knu-1 pCRC::iaaH* (**d**) fruits. Above: close-up views of fruit tips. Below: shapes of whole fruits. Arrowheads indicate stigma structures. Two stigma-like structures were observed in the mock-treated *crc-1 pCRC::iaaH* fruits due to carpel separation, while IAM treatment led to carpel fusion. **e–h** Longitudinal section of mock-treated *crc-1 pCRC::iaaH* (**e**), IAM-treated *crc-1 pCRC::iaaH* (**f**), mock-treated *crc-1 knu-1 pCRC::iaaH* (**g**), and IAM-treated *crc-1 knu-1 pCRC::iaaH* (**h**) fruits. Asterisks indicate carpels. **i–l** Scanning electron micrograph of mock-treated *crc-1 pCRC::iaaH* (**i**), IAM-treated *crc-1 pCRC::iaaH* (**j**), mock-treated *crc-1 knu-1 pCRC::iaaH* (**k**), and IAM-treated *crc-1 knu-1 pCRC::iaaH* (**l**) fruits. **m** Quantification of mutant phenotypes. Although IAM-treated plants had shorter styles than the controls, we categorized the mutants based on the presence or absence of two stigmatic structures. *p*-values were calculated by $\chi^2$ test ($n = 30$). **n** mRNA abundance of the stem cell marker *WUS*, in *crc-1 pCRC::iaaH* and *crc-1 knu-1 pCRC::iaaH* flowers with or without IAM. Bars = 1 cm in **a–d** bottom; 500 μm in **e–h**; and 1 mm in **i–l**

(Fig. 4b, c and Supplementary Fig. 9g), this ectopic expression could be due to the presence of multiple transcriptional activators of *TRN2*[25], feedback regulation via the auxin pathway, and/or lack of mobile signals from YABBY gene activities[46]. Moreover, dexamethasone (DEX) activation of biologically functional and glucocorticoid receptor (GR)-tagged CR protein led to a rapid

and reproducible decrease in *TRN2* expression (Fig. 4a and Supplementary Fig. 10a). These results suggest that CRC represses *TRN2* expression in stage 6 floral buds.

We next attempted to identify *cis*-regulatory elements important for *TRN2* expression. By performing phylogenetic shadowing in four different *Brassicaceae* species, we identified

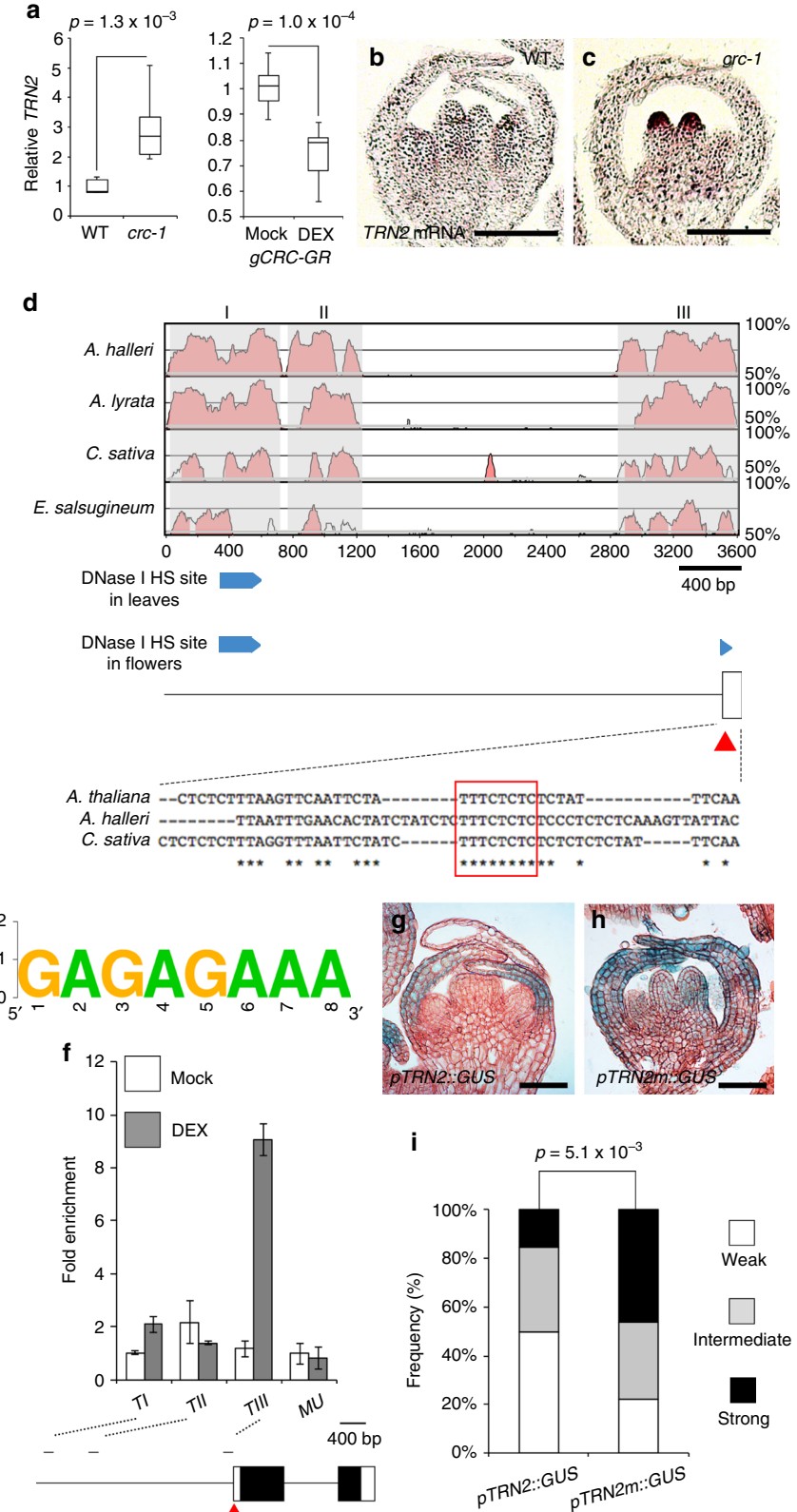

three conserved regions in the *TRN2* promoter (Fig. 4d). Among these regions, region I and III overlap with DNase hypersensitive sites, where transcription factors can bind[47]. Region III contains a DNase I hypersensitive site only in flowers, where CRC is specifically expressed (Fig. 4d and Supplementary Fig. 10b). Closer inspection of region III revealed the presence of a previously identified potential YABBY-binding site[48] (GA[A/G] AGAAA) (Fig. 4d, e and Supplementary Fig. 10c). Chromatin immunoprecipitation (ChIP) analysis using a Myc epitope-tagged CRC protein and floral tissues from flowers that were synchronized in development showed that CRC was specifically enriched at the region TIII containing a conserved site (Fig. 4f). This result suggests that CRC specifically binds to region III of the *TRN2* promoter to repress its transcription in a flower-specific manner, although *TRN2* expression in leaves or vasculature is likely regulated by other transcription factor(s) bound by region I of the *TRN2* promoter.

To investigate whether the YABBY-binding site is necessary for *TRN2* expression, we mutagenized this site in the *TRN2* promoter, yielding *pTRN2m* (Fig. 4g–i and Supplementary Fig. 10d–g). We did not detect the β-glucuronidase (GUS) activity in developing carpels at stage 6 and the reporter was misexpressed when introduced into *crc-1* (Fig. 4g and Supplementary Fig. 10h, i). *pTRN2m::GUS* was continuously expressed throughout two carpels of stage 6 flowers (Fig. 4g, h). Increases in *pTRN2m::GUS* signals were also detected in sepals, most likely due to the activity of other YABBY family proteins (Fig. 4g, h). We analyzed the reporter constructs in an independent population of T1 plants ($n \geq 25$) to minimize possible positional effects on transgene expression. Significantly more *pTRN2m::GUS* lines had stronger signals than *pTRN2::GUS* lines ($p = 0.005$) (Fig. 4i). These results suggest that a conserved YABBY-binding site is required for proper repression of *TRN2* expression.

### *trn2* mutation partially suppresses *crc knu*.

To examine whether the floral meristem indeterminacy phenotype in *crc knu* is caused by the ectopic expression of *TRN2*, we eliminated *TRN2* activity in the *crc knu* double mutant (Fig. 5a–i). The *trn2-1* mutant showed reduced *TRN2* transcript levels with nonsense mutation[22] and had very small and twisted fruits (Fig. 5a, d, g and Supplementary Fig. 11). Two carpels were properly fused in the *trn2-1* mutant (Fig. 5a, d, g). The *trn2-1 crc-1 knu-1* mutant exhibited reduced indeterminacy compared to the *crc-1 knu-1* parental line ($p = 2.2 \times 10^{-3}$) (Fig. 5b, c, e, f, h, i, j).

We also investigated *WUS* expression in the triple mutant by both quantitative reverse transcription PCR (qRT-PCR) and in situ hybridization. Consistent with the phenotypic severity, removal of TRN2 activity from *crc knu* restored *WUS* mRNA levels to those of *trn2-1* (Fig. 5k). While no *WUS* mRNA was detected in stage 6 *trn2-1* flowers, *WUS* mRNA levels were very high in *crc-1 knu-1* flowers at the same stage. In the *trn2-1 crc-1 knu-1* triple mutant, compromised *WUS* levels were detected compared to *crc knu* (Fig. 5l–n). Taken together, these results indicate that *WUS* repression by CRC is partially dependent on the TRN2 activity.

### Misexpression of *TRN2* in *knu* mimics *crc knu*.

To determine whether *TRN2* misexpression is sufficient to mimic the *crc* mutant phenotype, we expressed *TRN2* under the control of the *CRC* promoter (*pCRC::TRN2*) (Fig. 6a–d). *pCRC::TRN2* fruits mimicked the *crc* mutant and no phenotypic difference was observed between *pCRC::TRN2* and the *crc-1* mutant ($p = 0.15$), while the *pCRC::TRN2* phenotype differed significantly from the wild type ($p = 4.3 \times 10^{-13}$) (Fig. 6a, b, e). We then examined the contribution of TRN2 to the *crc-1 knu-1* phenotype by transforming *knu-1* plants with *pCRC::TRN2*. In contrast to the *knu-1* parental line, the *pCRC::TRN2 knu-1* plants formed *crc knu* double mutant-like abnormal fruits bearing extra carpels and stamens due to *WUS* misexpression (Fig. 6b, c, e, f, h, j–l).

To investigate the link between ectopic *TRN2* expression and auxin homeostasis, we treated *pCRC::TRN2* lines with NPA. As observed in NPA-treated *crc-1* mutants, NPA-treated *pCRC:: TRN2* plants formed significantly more normal fruits than mock-treated plants ($p = 2.3 \times 10^{-7}$) (Supplementary Fig. 12a–d). Moreover, NPA treatment of *pCRC::TRN2 knu-1* plants significantly rescued the iteration phenotype ($p = 1.9 \times 10^{-10}$) (Fig. 6d, e). Unlike *pCRC::TRN2 knu-1* plants, NPA-treated *pCRC::TRN2 knu-1* plants produced fused carpels (Fig. 6c–i). The number of ectopic carpels that formed inside of fruits was reduced, as observed in the *knu-1* mutant (Fig. 6f, g and Supplementary Fig. 6c, d, g, h). In agreement with phenotypic results, qRT-PCR and in situ hybridization showed high levels of *WUS* expression in *pCRC::TRN2 knu* plants, but NPA treatment of these plants led to reduced *WUS* expression (Fig. 6j–m and Supplementary Fig. 12e). These results suggest that ectopic *TRN2* expression is sufficient to mimic the *crc* mutant phenotype.

### Plasma membrane-localized TRN2 modulates auxin homeostasis.

To investigate the function of TRN2, we examined the localization of this protein in reproductive tissues. The *gTRN2-GFP* transgene we generated rescued the *trn2* mutant phenotypes (Supplementary Fig. 13a–f). TRN2-GFP was mainly detected in the shoot apical meristem and vasculature (Fig. 7a). Consistent with the finding that *TRN2* encodes a transmembrane protein of the tetraspanin family, TRN2-GFP was localized to the plasma membrane (Fig. 7b–d and Supplementary Fig. 13g, h). We did not detect TRN2-GFP signals in stage 6 carpels, which mimicked the *TRN2* promoter expression pattern (Figs. 4b, g and 7e). Compared to the wild type, TRN2-GFP protein levels were elevated in the carpel primordia of the *crc* floral buds at stage 6 (Fig. 7f).

We next examined PIN1-GFP expression and localization in the *trn2* mutant background. In the wild type, PIN1-GFP was highly expressed in the medial domain of the gynoecium (Fig. 7g).

**Fig. 4** *TRN2* expression is repressed by CRC via a conserved YABBY-binding site. **a** *TRN2* mRNA levels in wild-type and *crc-1* flowers (left) and mock- and DEX-treated *gCRC-GR* (right). *p*-values were calculated by Student's *t*-test ($n = 3$). **b, c** *TRN2* mRNA in situ hybridization in longitudinal sections of wild-type (**b**) and *crc-1* (**c**) floral buds at developmental stage 6. **d** Top: pairwise alignment using mVISTA of the upstream intergenic regions of *TRN2* from four *Brassicaceae* species. I, II, and III are conserved regions identified in the *TRN2* promoter. Middle: *TRN2* promoter structure. Blue bars indicate significant DNase I hypersensitive regions in flowers. Significant regions of DNase I hypersensitive sites were obtained though the Integrated Genome Browser. Bottom: Sequence conservation in the III region of the *TRN2* promoter. **e** Weblogo of YABBY-binding site. **f** Anti-Myc ChIP in *ap1-1 cal-1 35S::AP1-GR* after synchronization of flower stages. PCR fragments (TI, TII and TIII, which contain conserved regions) and *TRN2* promoter diagram are shown below the ChIP data. Red triangles indicate conserved YABBY-binding site. **g, h** Reporter expression from wild-type (**g**) and YABBY-binding site-mutated *TRN2* promoter (**h**). GUS expression in stage 6 floral buds under the control of the *TRN2* promoter with **g** or without **h** the YABBY-binding site. **i** Visual scoring of *pTRN2:: GUS* or *pTRN2m::GUS* staining in the T1 population; $n > 32$ for each construct tested. *p*-values were calculated by Student's *t*-test. Bars = 100 μm in **b, c**; and 50 μm in **g, h**

By contrast, PIN1-GFP expression was reduced in *trn2*, while the PIN1-GFP expression domain was expanded in *crc-1*, in which TRN2 is ectopically expressed (Figs. 2j, k and 7g, h). This reduction was not gynoecium-specific, as PIN1-GFP expression

in *trn2* inflorescences was lower than in the wild type (Supplementary Fig. 13i, j). We also quantified the transport of radiolabeled IAA in the *trn2* mutant grown on MS plates (Supplementary Fig. 13k–n). This assay showed that *trn2* had reduced IAA transport relative to the wild type (Fig. 7i) ($p = 7.6 \times 10^{-7}$).

Although it was important to ascertain possible roles of TRN2 in auxin transport using *trn2* mutants, changes might be simply due to tissue composition issues. To further examine the role of TRN2 in auxin maxima, we compared the *DR5* levels in wild-type and *35S::TRN2* cells, including both *Arabidopsis* T87 cells and tobacco BY-2 cells. We did not observe any differences in terms of cell shape or morphology even if *TRN2* was overexpressed in either cell. Regardless of the cells examined, signals from the *DR5rev::3xVENUS* were reduced in *35S::TRN2* compared to the wild type (Fig. 7j, k and Supplementary Fig. 13o, p). We also performed immunostaining with auxin antibody using *Arabidopsis* protoplasts, which allows free auxin to be visualized[49–51]. Based on immunostaining, we detected fewer auxin foci in *35S::TRN2* protoplasts than in untransformed wild-type cells (Fig. 7l, m). Finally, misexpression of *TRN2* under the control of the *CRC* promoter led to reduced *DR5* signals in the medial domain of the gynoecium as seen in *crc* mutants (Fig. 7n, o). Taken together, our results suggest that TRN2 modulates auxin homeostasis.

## Discussion

Here, we show that the direct AG target CRC controls auxin homeostasis in the developing gynoecium through the negative regulation of *TRN2*, thus preventing the overgrowth of floral meristems when *KNU* is mutated (Fig. 7p). Two direct AG targets, *CRC* and *KNU*, synergistically regulate floral meristem termination, and mutations in these two genes, which act downstream of AG, disrupt the floral meristem determinacy. The confirmation of the striking, synergistic mutant phenotype observed in *crc knu* sheds light on the importance of these key genes. Synergistic effects triggered by two non-homologous genes are thought to be caused by disruption of multiple or different pathways[52]. Furthermore, strong floral meristem indeterminacy in single *crc* ortholog mutants has been reported in several other angiosperms[40]. The function of CRC identified in the current study may be conserved during evolution.

The AG-WUS and AG-KNU-WUS pathways play central roles in determining the timing of floral meristem termination[13,18,19]. Precisely how AG alters the activity of a restricted floral stem cell population was previously unknown. Our findings reveal a molecular framework for the spatial regulation of the floral meristem. CRC controls auxin homeostasis partially through *TRN2* repression and generates proper auxin maxima to reduce the activity of the floral meristem for subsequent gynoecium formation. In wild-type floral buds at stage 6, auxin maxima as visualized by *DR5* expression were observed in the medial domain of the gynoecium. In this domain, auxin maxima are established via regulation of auxin homeostasis, such as auxin transport and local auxin biosynthesis[43]. In *crc* or *crc knu* floral buds at stage 6, *DR5* expression was greatly reduced. When auxin transport was blocked by auxin transport inhibitors, *DR5* expression was recovered in *crc* or *crc knu* flowers. Not only NPA treatment, but also local auxin production by *iaaH* rescued the short style defects in *crc* and restored the phenotype of the *crc knu* double mutant to that of the *knu* mutant in terms of floral meristem determinacy (Supplementary Figs. 5 and 7). Since we could not identify any conditions that rescued one of these defects, we favor a model in which changes in auxin homeostasis in the medial regions (including tip of carpels) of the developing gynoecium cause both

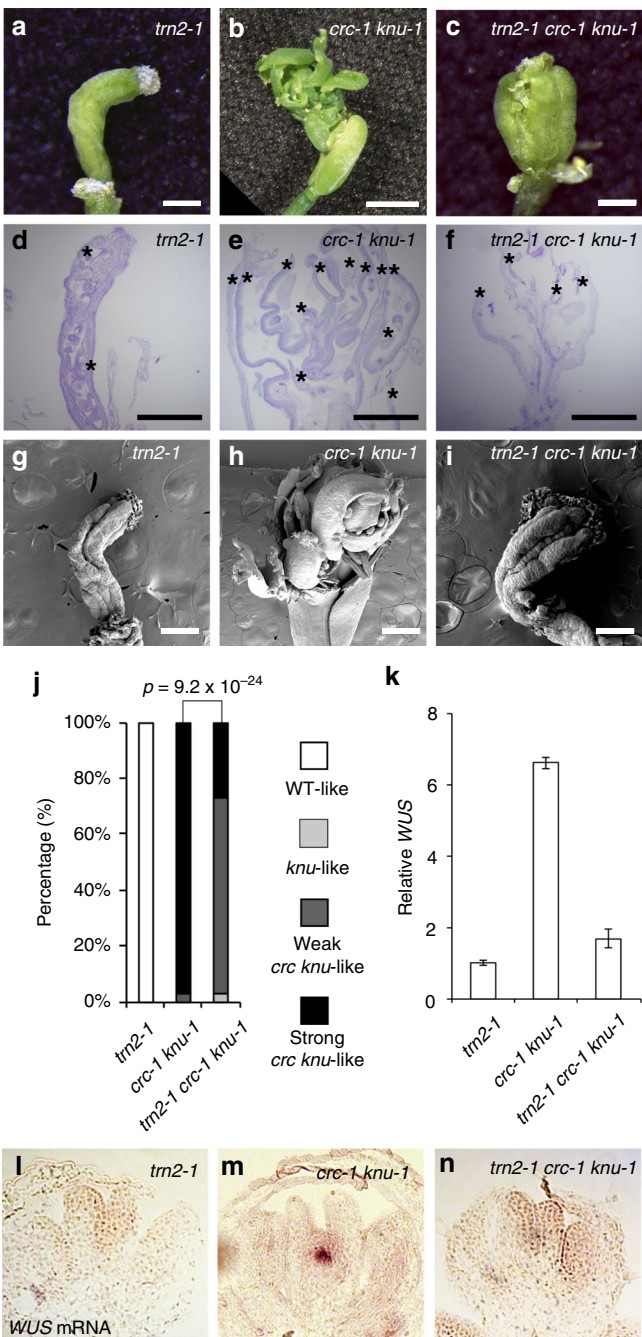

**Fig. 5** Removal of TRN2 activity from *crc knu* partially rescues the floral meristem termination defects in *crc knu*. **a–c** Morphology of *trn2-1* (**a**), *crc-1 knu-1* (**b**), and *trn2-1 crc-1 knu-1* (**c**) fruits. **d–f** Longitudinal sections of *trn2-1* (**d**), *crc-1 knu-1* (**e**), and *trn2-1 crc-1 knu-1* (**f**) fruits. Asterisks indicate carpels. **g–i** Scanning electron micrographs of *trn2-1* (**g**), *crc-1 knu-1* (**h**), and *trn2-1 crc-1 knu-1* (**i**) fruits. **j** Quantification of mutant phenotypes. *p*-values were calculated by $\chi^2$ test (*n* = 30). **k** mRNA abundance of the stem cell marker *WUS* in *trn2-1*, *crc-1 knu-1*, and *trn2-1 crc-1 knu-1* flowers. **l–n** *WUS* mRNA in longitudinal sections of *trn2-1* (**l**), *crc-1 knu-1* (**m**), and *trn2-1 crc-1 knu-1* (**n**) floral buds at developmental stage 6. Images are shown at the same magnification. Bars = 500 µm in **a**, **c**, **d–f**, **g–i**; 3 mm in **b**; and 100 µm in **l–n**

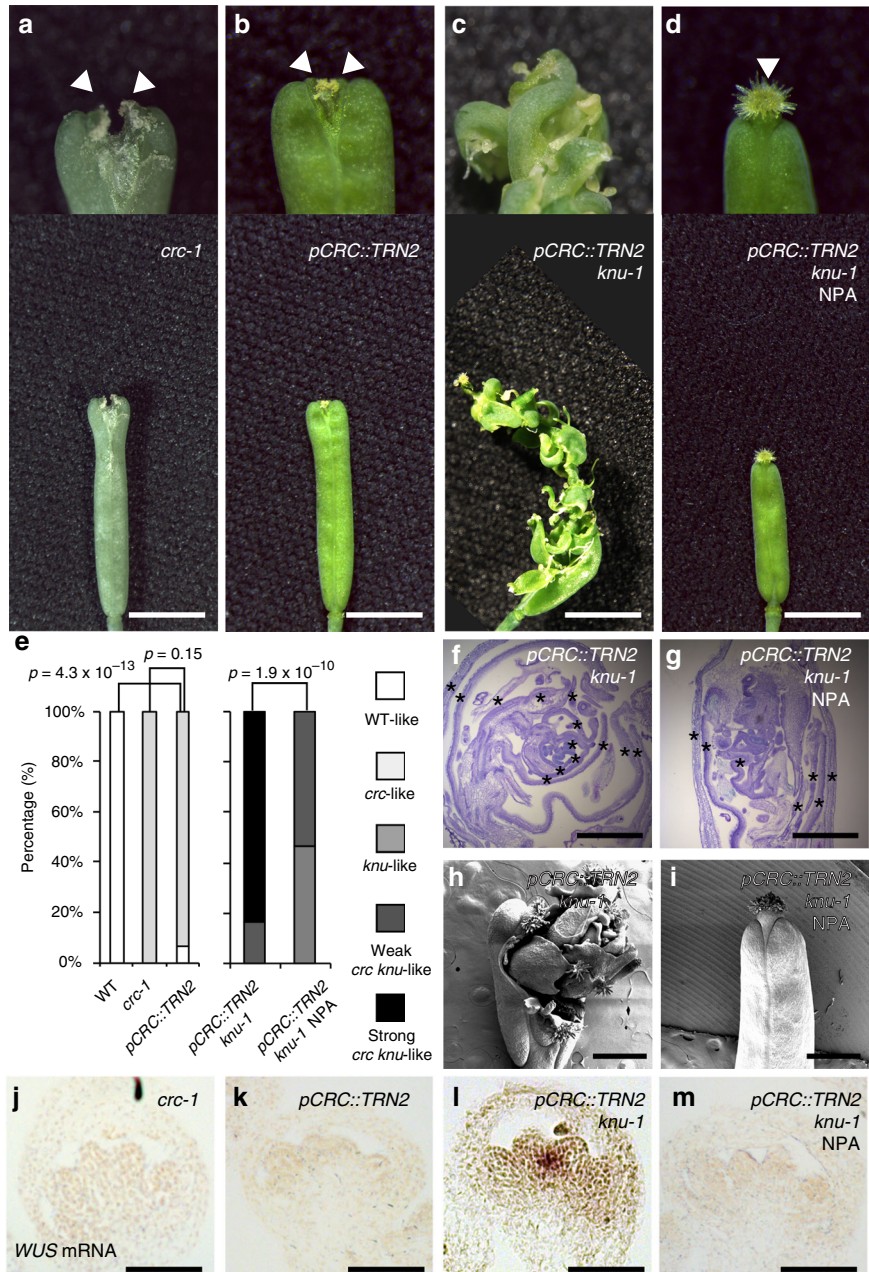

**Fig. 6** Ectopic expression of *TRN2* mimics the *crc* mutant phenotype. **a–d** Morphology of *crc-1* (**a**), *pCRC::TRN2* (**b**), *pCRC::TRN2 knu-1* (**c**), and NPA-treated *pCRC::TRN2 knu-1* (**d**) fruits. Top: close-up views of fruit tips. Bottom: shapes of whole fruits. Arrowheads indicate stigma structures. Two stigma-like structures were observed in the *crc-1* mutant as well as *pCRC::TRN2* due to carpel separation. Images are shown at the same magnification. **e** Quantification of mutant phenotypes. *p*-values were calculated by $\chi^2$ test (*n* = 30). **f**, **g** Longitudinal sections of mock-treated *pCRC::TRN2 knu-1* (**f**) and NPA-treated *pCRC::TRN2 knu-1* (**g**) fruits. Asterisks indicate carpels. **h**, **i** Scanning electron micrograph of *pCRC::TRN2 knu-1* fruits with reiterations of floral organs. *pCRC::TRN2 knu-1* (**h**) and NPA-treated *pCRC::TRN2 knu-1* (**i**) fruits. **j–m** Floral meristem activity marker, *WUS* mRNA, in longitudinal sections of *crc-1* (**j**), *pCRC::TRN2* (**k**), *pCRC::TRN2 knu-1* (**l**), and NPA-treated *pCRC::TRN2 knu-1* (**m**) floral buds at developmental stage 6. Images are shown at the same magnification. Bars = 1 cm in **a–d**; 500 μm in **f**, **g**; 1 mm in **h**, **i**; and 50 μm in **j–m**

style and floral meristem determinacy defects. CRC-mediated medial domain initiation through auxin homeostasis during stage 6 of flower development may both turn off floral meristem maintenance and trigger medial domain differentiation, resulting in the formation of the style and the normal tip. Although the pathway that functions downstream of auxin to repress *WUS* expression is not yet fully understood, one such transcription factor in this pathway has been identified. AUXIN RESPONSE FACTOR3 (ARF3)-mediated auxin perception controls numerous auxin-related events including gynoecium formation[53–56]

ARF3 directly binds to the proximal region of the *WUS* promoter to repress its expression[57]. Thus, *ARF3* might act downstream of the CRC-TRN2 pathway (Fig. 7p).

For decades, molecular and genetic approaches have been used to identify factors that participate in gynoecium development[12]. Numerous tissue-specific transcription factors have been identified[58]. Despite the critical importance of gene regulatory networks regulated by these factors, the downstream target(s) that actually plays a key role in gynoecium formation is currently unknown. Auxin controls gynoecium development[7], but the

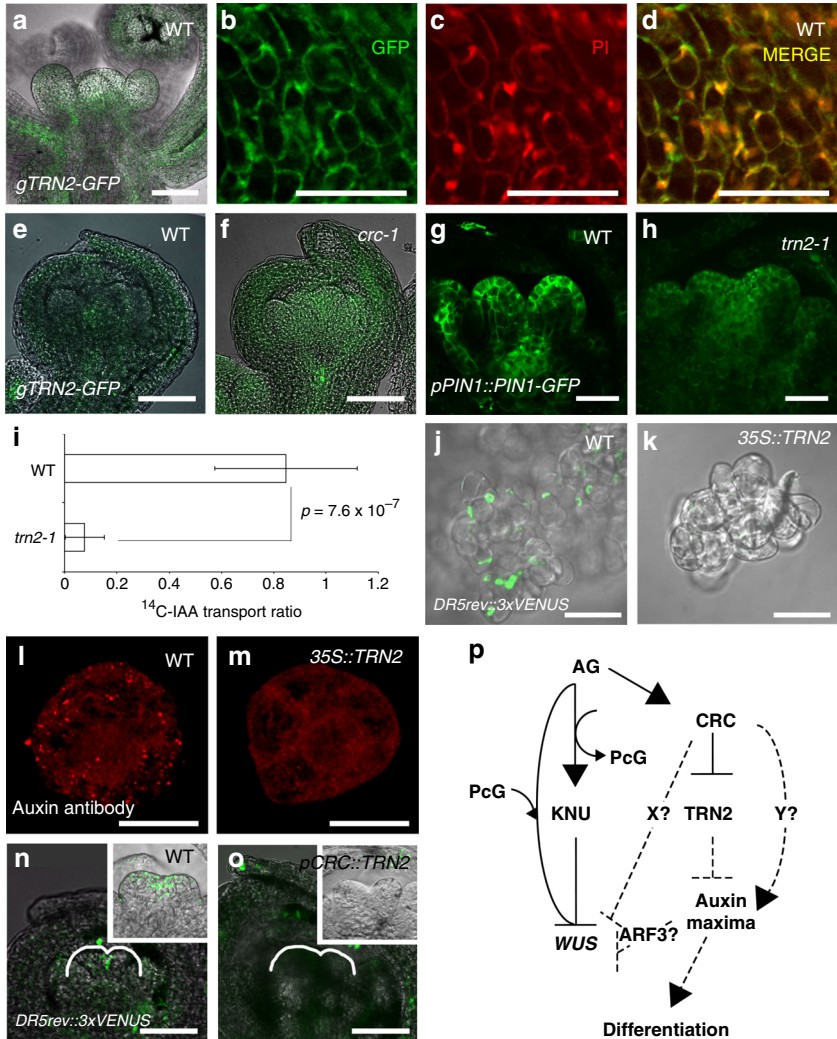

**Fig. 7** TRN2 is localized to the plasma membrane and regulates auxin homeostasis. **a** *gTRN2-GFP* expression in a wild-type inflorescence. **b–d** *gTRN2-GFP* expression (**b**), propidium iodide staining (**c**), and merged (**d**) images of cells in lower part of stage 2 floral primordia. **e**, **f** *gTRN2-GFP* expression in stage 6 floral buds of the wild type (**e**) and *crc-1* (**f**). **g**, **h**, *pPIN1::PIN1-GFP* expression in wild-type (**g**) and *trn2-1* (**h**) floral buds. **i** Basipetal IAA transport detection in wild-type and *trn2* stem. *p*-values were calculated by Student's *t*-test (n = 12 for WT, n = 14 for *trn2* mutants). **j**, **k** *DR5rev::3xVENUS* expression in wild-type (**j**) and *35S::TRN2*-expressing (**k**) *Arabidopsis* T87 cells. **l**, **m** Immunostaining of auxin in wild-type (**l**) and *35S::TRN2*-expressing **m** *Arabidopsis* T87 cells. **n**, **o** *DR5rev::3xVENUS* expression in wild-type (**n**) and *pCRC::TRN2* (**o**) floral buds at stage 6. Insets are closer inspection of floral buds. **p** Current model for AG-mediated floral meristem termination. Bars = 50 μm in **a**, **e**, **f**, **n**, **o**; 25 μm in **g**, **h**, **j**, **k**; and 10 μm in **l**, **m**

genes involved in auxin homeostasis are often expressed throughout the entire plant. Thus, auxin homeostasis must be fine-tuned in a tissue- or stage-specific manner. This study uncovers a molecular link between tissue-specific transcription factors[58] and a gene controlling auxin maxima. AG-activated CRC represses *TRN2*, which at least partially controls floral meristem termination. *TRN2* expression is correlated with the PIN1 reporter in the carpel primordia at stage 6 of flower development. Both TRN2 and PIN1 proteins were misexpressed in the *crc* mutant (Figs. 2j, k and 7e, f). In contrast, *trn2-1* showed reduced PIN1 expression with compromised auxin transport activity (Fig. 7g–i). Future investigations are needed to understand how TRN2 controls auxin homeostasis. Although the *crc* mutant phenotype is only seen in flowers, the *trn2* mutant shows a pleiotropic phenotype, such as twisted roots or leaves. Thus, CRC seems to act as a flower-specific repressor rather than a general repressor of *TRN2* throughout the plant body. Furthermore, *TRN2* was misexpressed throughout the carpel, while *CRC* is only expressed in the abaxial side of carpels. This misexpression

could be due to feedback regulation through auxin homeostasis. Alternatives are lack of mobile signals derived from CRC activity, as non-cell-autonomous activities have been reported for all characterized YABBY genes[46]. Since ETHYLENE RESPONSE FACTOR 115[59] and PSEUDO-RESPONSE REGUALATOR 5[60] are expressed in carpels[61] and positively regulate *TRN2*[25] expression, those activators and CRC might act antagonistically to control *TRN2* expression.

Since introducing the *trn2* mutation into the *crc knu* double mutant led to only partial phenotypic rescue, *TRN2* is not clearly the sole downstream target of CRC in the control of floral meristem termination. Such target(s) might be among the 365 misexpressed genes in *crc knu* compared to *knu*. In particular, the final six CRC candidate genes, which are most likely involved in floral meristem termination included *SHEPHERD* (*SHD*) and *CURLY LEAF* (*CLF*). Treatment with the auxin transport inhibitor rescued *crc knu* mutants more thoroughly than did introduction of the *trn2* mutation. The different degree of rescue observed in the triple mutants suggests that an auxin-related

target might exist that acts directly downstream of CRC and in parallel with TRN2 (Figs. 5 and 7). Because our GO term analysis identified many genes involved in metabolic processes (Fig. 2), it is also possible that CRC has more downstream targets that jointly control auxin transport and metabolism (biosynthesis, conjugation, and degradation) at multiple levels. Further introducing mutations in CRC candidate target genes into the *trn2 crc knu* triple mutant would provide insight into the importance of other CRC targets and the CRC downstream pathways for floral meristem determinacy.

## Methods

**Genetic stocks and growth conditions**. All *Arabidopsis thaliana* seed stocks were in the Landsberg erecta (L*er*) background or backcrossed into L*er* three to four times. The *crc-1*[26], *knu-1*[17], *wus-1*[14], *ag-1 35S::AG-GR*[62], *ap1-1 cal-5 35S::AP1-GR*[63], *pPIN1::PIN1-GFP*[41], and *DR5rev::3xVENUS*[42] lines were used in this study. For mature plant materials, seeds were sown in pots containing vermiculite and Metro-Mix. For antibiotic selection, plants were grown on Murashige and Skoog (MS) plates. Culture medium consisted of half-strength Murashige and Skoog (MS) salts and 0.8% agar, pH 5.6. All plants were grown at 22 °C under 24-hour light conditions.

**Phenotypic and statistical analyses**. The first five fruits from six individual plants (total of 30 fruits) for the wild type and each mutant were categorized into two to five classes based on phenotypic strength. The first five fruits from twenty individual plants (total of 100 fruits) were characterized for Fig. 5j. Plants to be directly compared were grown side-by-side at the same density per pot to minimize potential micro-environmental differences in the growth chamber. Representative images were photographed under an AXIO Zoom.V16 (ZEISS) microscope. Statistical significance was computed using a $\chi^2$ test.

**Cloning and transformation**. To construct *gCRC-GFP*, *gCRC-Myc*, and *gCRC-GR*, a genomic fragment covering the 3492 bp region upstream of the *CRC* translation start site and the 1204 bp coding region were amplified from Landsberg erecta genomic DNA. The resulting DNA fragments were cloned into the binary vectors pGreen0311 (for mGFP) and pGreen0280 (for Myc and GR) and sequenced (http://www.pgreen.ac.uk). To construct *pCRC::TRN2* or *pCRC::iaaH*, the 3.5 kbp *CRC* promoter and coding sequence of *TRN2* or *iaaH* were sub-cloned into pGreen0311 and sequenced. To amplify the *iaaH* coding sequence, DNA extracted from transgenic plants containing this gene was used as a template. To construct *gTRN2-GFP*, a genomic fragment covering the 4.1 kb region upstream of the *TRN2* translation start site and its coding region was amplified from genomic DNA and cloned into vector pGreen0311. After cloning of these DNA fragments into the pGreen vector, whole functional fragments were transferred into the pENTR 1002 entry vector for the LR reaction; pBGW and pKGW (Invitrogen) were used as destination vectors for the LR reaction.

To construct *pTRN2::GUS*, the 4.1 kbp *TRN2* promoter was amplified using *gTRN2-GFP* as a PCR template and sub-cloned into pENTR/D-TOPO. To mutate the CRC-binding site in the *pTRN2* promoter, site-directed mutagenesis was performed using a PrimeSTAR Mutagenesis Basal Kit (Takara) to obtain *pTRN2m*-pENTR/D-TOPO. After sequencing, the resulting *pTRN2* fragments were Gateway-cloned into pGWB3[64] by the LR reaction.

For plant transformation, the inflorescences were dipped in *Agrobacterium tumefaciens* for 2 min and incubated for 1 day in humid conditions at 4 °C[65]. T1 seeds were collected and screened for antibiotic resistance. More than 20 T1 plants were screened, and representative lines were chosen for further characterization. Primers used for cloning are listed in Supplementary Table 2.

**Chemical treatment**. For DEX treatment, the inflorescences were sprayed once using 10 μM DEX with 0.01% Silwet for 1 min. The DEX (Sigma) was dissolved in 100% EtOH. Control mock treatment was performed using equal amounts of EtOH with 0.01% Silwet. For RT-PCR, RNA from *ag-1 35S::AG-GR* and *crc-1 gCRC-GR* was extracted at 2 and 4 h and only 4 h after DEX treatment, respectively. For ChIP using *ap1-1 cal-1 35S::AP-GR*, or *ap1-1 cal-1 35S::AP1-GR gCRC-myc*, the ChIP tissues were fixed 4 days after DEX treatment.

For NPA (Tokyo Chemical Industry), 1-NOA (Sigma), PCIB (Sigma), IAA (Wako), NAA (Wako), and 2,4-D (Wako) treatment, 30 μL of 100 μM drugs with 0.01% Silwet was dropped onto the inflorescences of *crc-1*, *crc-1 knu-1*, *pCRC::TRN2*, and *pCRC::TRN2 knu-1* plants once when the plant height was ~2 cm. At 6–10 days after treatment, fruit phenotypes were observed. Since all the drugs were dissolved in DMSO, DMSO was used as a mock treatment control.

For IAM (Tokyo Chemical Industry) treatment, 10 μM IAM with 0.015% Silwet was dropped onto the inflorescences of *pCRC::iaaH crc* or *pCRC::iaaH crc knu* plants for 3 consecutive days when the plant height was ~2 cm. At 6–10 days after treatment, fruit phenotypes were observed. The IAM was dissolved in DMSO. Control mock treatment was performed using equal amounts of DMSO with 0.015% Silwet.

**Scanning electron microscope**. To observe the samples by scanning electron microscopy (SEM), tissues were fixed in FAA (45% EtOH, 5% formaldehyde, and 5% acetic acid) overnight at room temperature and dehydrated through a graded ethanol series and acetone. The tissues were critical-point-dried with liquid $CO_2$ in an HCP-2 critical point dryer (Hitachi) and gold-coated with E-1010 (Hitachi) before SEM imaging. The tissues were imaged under an S-4700 (Hitachi) with an accelerating voltage of 15 kV. More than 10 fruits were observed, and representative images are shown.

**Sectioning**. For sectioning, tissues were embedded in Technovit 7100 resin (Heraeus) and polymerized at room temperature overnight. Sections (8 μm thick) were cut with an RM2255 microtome (Leica), dried, stained with 0.05% Toluidine Blue (Wako), mounted on a microscope slide with one or two drops of Mount-Quick (Daido Sangyo), and imaged using an Axio Scope A1 microscope (ZEISS). More than five fruits were sectioned, and representative images are shown.

**In situ hybridization**. For in situ hybridization, inflorescences <5 cm long were harvested and fixed in FAA. The resulting tissues were dehydrated in a graded ethanol series, replaced with xylene, and embedded in Paraplast plus (Sigma). Sections (8 μm thick) to be directly compared were processed together. The full length, 5′-UTR and first 472, and 921 bp were used to generate *WUS*[66], *AGO10*[67], and *PHB*[68] probes, respectively. The *CRC* and *TRN2* probes were amplified using cDNA prepared from L*er* and cloned into the pCRII vector (Invitrogen). In vitro transcription was conducted by T3, T7, or SP6 polymerases. Hybridized tissues were mounted on a microscope slide with one or two drops of Mount-Quick (Daido Sangyo) and observed under an Axio Scope A1 microscope (Zeiss). Two independent experiments were performed, and similar results were obtained. See Supplementary Table 2 for the primers used for in situ probe cloning.

**GFP observation**. For GFP observation, confocal laser scanning microscopy (FV1000: Olympus) with objective lens (UPlanSApo: Olympus) was used. Plants 4–8 cm tall that were directly compared were grown side-by-side at the same density per pot to minimize potential micro-environmental differences in the growth chamber. To produce sections of inflorescences, tissues were embedded into 5% agar and sliced with a Liner Slicer PRO7 vibratome (Dosaka). The resulting flower sections were placed onto glasses, mounted on a microscope slide with one or two drops of water or 20 μg/ml PI (Sigma), and observed immediately with an FV1000. The same offset and gain settings were used for all plants for which signal intensity was directly compared. More than five flowers were observed under a confocal microscope, and representative images are shown. Three independent experiments for quantification were performed, and similar results were obtained. The data from one representative experiment are shown.

**Reverse transcription polymerase chain reaction**. For RT-PCR, total RNA from floral bud clusters up to stage 10 for mutants[8] or *ap1 cal* floral bud clusters at 4 h after DEX treatment was extracted using an RNeasy plant mini kit (Qiagen). To remove genomic DNA, an RNase-Free DNase set (Qiagen) was used. The cDNA was synthesized using PrimeScript RT Master Mix (Takara). The cDNA levels were determined with a Light Cycler 480 (Roche) and Light Cycler 480 release 1.5.1.62 SP software (Roche) using FastStart DNA essential DNA Green Master (Roche). The RT-PCR experiments were normalized using the internal control gene *EIF4* (At3g13290). The mean and s.e.m. were determined using at least four technical replicates from one representative experiment. Two or three independent experiments were performed, and similar results were obtained. See Supplementary Table 2 for qRT-PCR primers.

**GUS staining**. Plant tissue was fixed at room temperature in 90% ice-cold acetone for 15 min, rinsed in GUS staining buffer without 5-bromo-4-chloro-3-indolyl-b-D-glucuronide (X-Gluc), and stained with GUS staining solution (50 mM NaPO₄, 2 mM K₄Fe(CN)₆, 2 mM K₃Fe(CN)₆, 2 mM X-Gluc) at 37 °C overnight. The resulting stained tissues were incubated in 70% EtOH for more than 1 week. Plants to be directly compared were grown and stained at the same time. Signal strengths were categorized by visual scoring of independent T1 lines. More than 32 were tested for each construct. Statistical significance was calculated using a Student's *t*-test. Tissues were mounted on a microscope slide with one or two drops of chloral hydrate-based clearing solution or 0.05% neutral red (Wako), incubated at room temperature for ~1 h, and imaged using an Axio Scope A1 microscope (Zeiss), AxioCam ERc 5 s camera (Zeiss), and ZEN2 software (Zeiss).

**Chromatin immunoprecipitation assay**. To investigate the protein binding of stage-specific transcription factors, *ap1-1 cal-1 35S::AP1-GR* inflorescences were treated with 10 μM DEX and harvested 4 days later. Inflorescence tissues were fixed with 1% formaldehyde for 15 min, frozen with liquid nitrogen, and kept −80 °C until use. Chromatin was extracted and sonicated to produce DNA fragment under 500 bp. The sonicated DNA was pre-cleared with Protein A beads (Thermo Fisher Scientific) for 2 h at 4 °C. After centrifugation, antibody was added. The tubes were incubated overnight at 4 °C on a rotating device. Beads were then added and incubate for more than 6 h at 4 °C on a rotating device. The resulting beads were

washed twice with low salt buffer, high salt buffer, LiCl buffer, and TE buffer. DNA was eluted by incubation for 1 h at 65 °C. Protein-DNA crosslink reversal was conducted overnight at 65 °C. DNA was recovered using QIAquick PCR purification kit (Qiagen). The AG antibody[69] or Myc antibody (9E10; Santa Cruz) was used. The *MU* gene was used as the negative control locus for ChIP experiments. The ratio of ChIP over input DNA was calculated by comparing the reaction threshold cycle for each ChIP sample to a dilution series of the corresponding input sample and was normalized over mock control loci to obtain fold change. Three independent experiments were performed, and similar results were obtained. The data from one representative experiment are shown. See Supplementary Table 2 for ChIP primers.

**Gene expression profiling**. Microarray analysis with three biological replicates was performed with the NimbleGen $12 \times 135$ K arrays (Roche). The plants were grown at 22 °C under 24-h light conditions. When the plants reached a height of 5–10 cm, inflorescence bud clusters containing flowers of up to stage 10 were harvested. Gene expression was analyzed by Arraystar (DNASTAR) with gene annotations from TAIR7. Genes showing a 2.0-fold change in expression within a 99% confidence interval were considered to be differentially expressed and are presented in Supplementary Data 1. These genes might be involved in various CRC-mediated events, such as floral meristem determinacy, nectary formation, and gynoecium formation. GO term analysis was performed using agriGO software version 1.2 (http://bioinfo.cau.edu.cn/agriGO/). AGI codes from Supplementary Data 1 were used as queries, and all GO terms and hierarchical tree graphs of over-represented GO terms in biological process were downloaded from the agriGO website and are shown in Supplementary Data 2 and Supplementary Fig. 4b, respectively.

**Phylogenetic shadowing and motif analysis**. For phylogenetic shadowing, the *TRN2* promoter sequences from *Arabidopsis halleri* (BASO01001507.1), *Arabidopsis lyrata* (BASP01017915.1), *Cannabis sativa* (JFZQ01000232.1), and *Eutrema salsugineum* (AHIU01003814.1) were obtained by NCBI blastn using the *Arabidopsis thaliana TRN2* promoter sequence as a query (http://www.ncbi.nlm.nih.gov/genbank/). The promoter sequences were aligned by mVISTA (http://genome.lbl.gov/vista/mvista/submit.shtml). Closer inspection of promoter sequences was performed using CLUSTALW (http://www.genome.jp/tools/clustalw/). The sequence logo of cis-elements was generated by WebLogo (http://weblogo.berkeley.edu).

**Culture and transformation of *Arabidopsis* and tobacco cell lines**. Tobacco BY-2 and *Arabidopsis* MM2D cells were cultured in MS medium (4.4 g/L MS salt mix, 30 g/L sucrose, 0.2 g/L 2,4-Dichlorophenoxyacetic acid) at 22 °C in the dark. BY-2/MM2D cells grown on MS agar medium were inoculated into 10 mL of liquid medium and cultured on a rotary shaker at 100 rpm for 1 week. Agrobacterium cells (5 mL at OD600 = 0.8) were washed three times with 5 mL of MS medium (with 250 µg/mL Carbenicillin) and resuspended in 5 mL of MS medium. The 10 mL BY-2/MM2D cell cultures were combined with 100 µL of Agrobacteria in MS and cultured for ~36 h. The BY-2/MM2D cells were then washed four times with MS (with 250 µg/mL Carbenicillin), resuspended in 2 mL of MS, and plated onto MS agar medium. For *DR5rev::3xVENUS*, the pCR2.1-TOPO vector containing both *DR5rev* promoter and 3xVENUS-4xAlanine-N7 was used[42]. To generate the *35S::TRN2-GFP* construct, eGFP in pGreen with *35S::eGFP-mGFP* was replaced by the *TRN2* coding sequence. See Supplementary Table 2 for cloning primers. The *DR5rev::3xVENUS* and *35S::TRN2* constructs were transformed into T87 cells by *Agrobacterium tumefaciens* (strain C58C1) transformation. The VENUS signals were observed under a Zeiss LSM510 microscope. More than 10 cells were observed, and representative images are shown.

**Auxin transport assay**. For basipetal auxin transport assay, the basal 10 mm length of primary inflorescences was used. An inflorescence stem was inserted into a 1.5 ml tube that contained 30 µl of mineral medium and $^{14}$C-IAA (1.8 µM, 3.7 kBq/ml, American Radiolabeled Chemicals, Inc., St. Louis, MO) in the inverted orientation and incubated at 22 °C for 16 h in a humidity chamber. For quantification of $^{14}$C levels, the both apical and basal ends of the stem were cut at the length of 3 mm and were digested by Soluene 350 (Perkin Elmer, Waltham, MA) in capped vials at 50 °C for a day. After the addition of scintillation cocktail (Hionic Fluor, Perkin Elmer, Waltham, MA), the $^{14}$C levels were measured by a liquid scintillation counter (Aloka, Tokyo, Japan). To calculate ratio of basipetal IAA transport, the $^{14}$C levels of the basal side was divided by $^{14}$C levels of the apical side. More than twelve stems were observed for each genotype. Statistical difference was evaluated by Student's t-test.

**Immunostaining of auxin in BY-2 protoplasts**. To perform immunostaining, a few droplets of 400 mM mannitol were placed onto the leaves to protect the cells. The tips and petioles of the leaves were removed, and the leaves were cut into strips to facilitate digestion. To digest the cell walls, the leaf strips (or 500 µL of BY-2 cells) were incubated in 3 mL of enzyme solution at room temperature for 3 h with gentle shaking. After incubation, the cells were filtered through a 35–70 µm nylon mesh filter, mixed with 3 mL of pre-chilled W5 buffer, and centrifuged at 100 g for 1 min to collect the cells. The cells were washed once, resuspended in cold W5

buffer, and incubated for 30 min on ice. After incubation, the cells were centrifuged at 100 g for 1 min and resuspended in 300 µL of W6 buffer. A circle was drawn on ProbeOn glass slides with a PAP pen. The 300 µL cell suspension was dropped onto the circle, and the slide was incubated for 1 h to allow the cells to adhere to the slide. The W6 buffer was carefully removed, and the cells were fixed with fixing solution (4% formaldehyde in W6 solution) and incubated for 1 h at room temperature. The cells were washed three times with 300 µL of TSW buffer for 10 min per wash for permeabilization, followed by incubation with primary antibody[49–51] (anti-auxin) (Sigma-Aldrich) diluted in 300 µL of TSW buffer at 4 °C for 16 h in a moist chamber. The samples were washed three times with 300 µL of TSW buffer for 10 min per wash to remove the unbound antibody, followed by incubation with secondary antibody diluted in 300 µL of TSW buffer at room temperature for 3 h. After three additional washes with 300 µL of TSW buffer for 10 min per wash to remove the unbound secondary antibody, the samples were mounted with 70 µL of DAPI, covered with a coverslip, and sealed with clear nail polish for longer storage. The VENUS signals were observed under a Zeiss LSM510 microscope. The VENUS signals were observed under a Zeiss LSM510 microscope. More than 10 cells were observed, and representative images are shown.

**Data availability**. The data that support the findings of this study are available from corresponding author on request. The microarray data have been deposited in GEO under accession number GSE88969.

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

## Acknowledgements

We would like to thank Mitsuhiro Aida for technical advice on SEM; Masahiro Akiyama for real-time PCR; Yuri Tajima for ChIP; Muthi Ikawati for microtome sectioning; Yuki Kubota, Kei Hiruma, and Naoki Takahashi for confocal microscope; Hufumi Otani for sequencing; Akie Takahashi, Eiko Nakamoto, Kyoko Sunuma, and Taeko Kawakami for technical assistance; Sachi Ando, Jinfeng Wu, and Yuka Kadoya for checking the draft of this manuscript; Elliot M. Meyerowitz for *DR5rev::3xVENUS* seeds and construct; Xu Jian for unpublished transgenic plants containing the *iaaH* gene; and Takashi Hotta for helpful discussion. This work was supported by grants from Japan Science and Technology Agency 'Precursory Research for Embryonic Science and Technology (no. JPMJPR15QA)', JSPS KAKENHI Grant-in-Aid for Scientific Research on Innovative Areas (no. 16H01468), the NAIST foundation, the Sumitomo Foundation and the Takeda Science Foundation to N.Y., grants from Japan Science and Technology Agency 'Precursory Research for Embryonic Science and Technology (no. JPMJPR15Q7)' to K.T., and grants from the NAIST foundation, the Mitsubishi Foundation, JSPS KAKENHI Grant-in-Aid for Scientific Research on Innovative Areas (no. 15H01234, 15H01356, and

17H05843), JSPS KAKENHI Grant-in-Aid for Scientific Research A (no. 15H02405), Grant-in-Aid for challenging Exploratory Research (no. 15K14549), Temasek Life Sciences Laboratory, and the National Research Foundation Singapore under its Competitive Research Programme (CRP Award NRFCRP001-108) to T.I.

## Author contributions

N.Y., J.H. and T.I. conceived this study. N.Y. and J.H. performed all experiments except for microarray analysis and auxin transport assay. The microarray experiment was conducted by J.H. and analyzed by J.H., N.Y. and Y.X. The auxin transport assay was conducted by N.Y. and K.T.; N.Y. and T.I. wrote the paper with input from J.H., Y.X. and K.T. All authors read and approved the final version of the manuscript.

## Additional information

**Competing interests:** The authors declare no competing financial interests.

