## [Peer Review File · Nature Communications]

Reviewers' comments:

Reviewer #1 (Remarks to the Author):

This manuscript by Yamaguchi et al presents interesting work showing how a target of AG, CRC, is required for floral meristem termination and gynoecium formation through regulation of auxin dynamics. They show how CRC functions with KNU to control determinacy and that CRC directly represses a plasma-membrane protein-encoding gene TRN2 to control auxin dynamics.

The data in the manuscript are solid and provides interesting new knowledge of the transition between meristem maintenance and organ initiation, which will be of interest to a wide audience such as the readers of Nature Communication. In particular, the genetics presented gives unequivocal support to the role of CRC and KNU in repressing WUS expression to promote transition from floral meristem maintenance to gynoecium development. The paper is logically structured, although the writing could be improved. Especially the discussion is not sufficiently concise and will require significant revision. Please see a list of comments below.

1. In this work and elsewhere in the literature it is shown that CRC has a role in floral meristem termination as well as in gynoecium patterning and here the authors also convincingly demonstrate that CRC regulates auxin dynamics. However, from the experiments presented in this manuscript it is not clear that CRC controls floral meristem termination via regulation of auxin dynamics. In fact, the lack of DR5 signal at the apex of stage-6 *crc* gynoecia seems more related to the split style phenotype. Moreover, it is indeed the split style defect that is rescued upon NPA application and in the CRC-*iaaH* line. It therefore seems that the effect of CR on auxin is separate from the role of CRC in floral meristem termination with KNU. In my opinion, the authors should make this distinction clearer. This also includes revising the title of the manuscript, which makes the claim that it is the control of auxin transport that governs the floral meristem termination. To my mind that is not demonstrated in the manuscript.

2. The experiment using *iaaH* is elegant and clearly shows that auxin homeostasis is important in this process. However, on the last line of page 7, the authors specifically claim that these data uncover a role for CRC in auxin transport. This experiment does not show that and it therefore cannot be concluded at this stage whether it is transport, biosynthesis, degradation or signaling that is important.

3. The data in Fig. 7i-l are confusing in that the genetics and the model in Fig. 7m suggests that TRN2 function to inhibit PINs. However, the data in Fig. 7i-l seem to suggest that auxin is transported out of these callus cells and into the medium (how is PIN1-GFP localized in these cells?). This apparent contradiction of their model needs to be better explained.

4. The promoter analysis of TRN2 is clear and convincing showing that the YABBY element is important for control of TRN2 expression. Although, TRN2 expression by in situ in WT and *crc* mutant is presented in Fig. 4b,c, it would be a good to have the TRN2:GUS in *crc* for comparison to Fig. 4g,h.

5. ARF3 appears prominently in the Discussion although no data on ARF3 are included in the manuscript. However, the arguments for linking ARF3 to AG/CRC/KNU regulation by the authors are sound and based on evidence from the literature. Even so, the reference to similar phenotypes between *crc knu* and *arf3* is curious, since Supplementary Fig. 7 only shows PIN1-GFP and in situ (Supplementary Fig. 13 doesn't seem to exist). It is impossible from these images to make the comparison that the authors do. Also, given the connection between auxin and ARF3, the authors should include a reference to the recent work by Simonini et al. (2016) Genes Dev on the mechanism

of ARF3 in auxin signaling.

6. Following on from the point above, the writing of the Discussion could be significantly improved. As is, it does not provide a clear summary and discussion of the data. It seems messy and unfounded claims/speculations are made. For example, the suggestion that the robustness of the redundant CRC/KNU pathways may exist to promote adaptation to environmental change is not based on any experiments, although it could easily be tested if *crc* or *knu* mutants shows altered responses to e.g. higher temperature. There is also a statement that NPA treatment fully rescues the *crc knu* double mutant. This is not correct; it merely brings it back to the *knu* phenotype.

Minor comments

1. It is mentioned three times on first page of the Introduction that the female reproductive structure is known as the gynoecium. One should be enough.

2. On page 4 in Introduction, auxin transport and PIN proteins are described with a reference to Okada et al. 1991 (ref. 21). It would be more appropriate to refer to the Gälweiler et al (1998) Science paper where PIN1 is cloned.

3. There are issues with reference numbering. For example, on page 6 ref 39 is used to refer to metabolic pathways producing hormones. This should be ref 40. Also in the second line of the next paragraph. Ref 40 is used to refer to NPA treatment. This should surely be ref 32. In addition, on page 11 ref 32 is used to refer to *crc* ortholog mutants in other angiosperms, although ref 32 contains no such information. Perhaps the authors meant to refer to Orashakova et al. (2009) Plant J 58: 682-693? There may be other reference mistakes that I haven't spotted.

4. On page 6, the authors state they performed an expression analysis of the pPIN1::PIN1-GFP marker line. However, it is not just expression, it is also PIN1-GFP localization.

4. In Figure 4d, four Brassicaceae species are used in a Phylogenetic shadowing. In Materials and methods *C. sativa* is listed as *Cannabis sativa*, which is part of the Cannabaceae family. I assume the authors meant to write *Camelina sativa*.

5. There is a reference to a Supplementary Figure 13 on page 12. This figure was not included in the material I received.

6. On page 9, there is a description of the morphological defect of the *trn2-1* mutant with a reference to panels in Fig. 5. However, the reference to Supplementary Fig. 9 (showing qPCR data of TRN2 expression in WT and *trn2-1*) does not match the text. The expression data should therefore either be mentioned or the figure removed.

Reviewer #2 (Remarks to the Author):

Yamaguchi et al. aim to provide a direct molecular link between gynoecium establishment, auxin transport and floral meristem determination. Based on previous research, floral meristem determination linked with WUS repression has been shown to be controlled by several pathways. Until now, only the direct and KNU-mediated repression of WUS by the floral homeotic regulator AG has been reported, while mutant phenotypes suggest additional pathways. Using genetic analyses, the authors show that the known direct AG-target CRC contributes to meristem determinacy and WUS repression additionally to KNU. In gene expression microarray experiments comparing *knu-1* single

mutants and *knu-1 crc-1* mutants, the authors identify 363 significantly different expressed genes. Among those are six genes known to be involved in meristem functions based on GO annotation. For example, *CURLY LEAF* and *SHEPHERD* are known to be involved in suppression of *WUS* activity. Nevertheless, the authors choose to study one gene, *TRN2*, and its potential role in auxin signaling as candidate pathway by which *CRC* mediates *WUS* repression in a non-cell autonomous manner. Treatment of *crc-1* and *knu-1* mutants with NPA suggest that auxin transport is indeed required for mediating *WUS* repression by *CRC*. *PIN1* localization however seems to be largely unaffected, besides a mild lateral expansion in the *crc* mutant. The authors further show that local overproduction of auxin in the *CRC* expression domain partially rescues the *crc knu* floral meristem termination defect, and a mutation in *trn2* partly suppresses the *crc knu* meristem termination defect. Finally, the authors provide evidence for a role of *TRN2* in auxin signaling.

Overall, the manuscript uses a balanced combination of alternative approaches to elucidate the challenging question on dissecting cellular pathways mediating *WUS* repression in the floral meristem. However, I have a number of major – especially technical – concerns and questions.

Major comments:

1. Page 6: The reasoning by which the authors initially choose to focus only on *TRN2* as potentially mediating the role of *CRC* in *WUS* repression is not well supported in my point of view. For example, *SHEPHERD* plays a role in spatial restriction of *WUS* expression (via the *CLV1* pathway). Among the 363 differentially expressed genes, there might well be more (yet uncharacterized) genes that are potential candidates directly downstream of *CRC*.
2. The role of *TRN2* in auxin transport/signaling is far from understood. The pleiotropic mutant phenotype suggests as well auxin-independent roles, and to my knowledge the current idea is that *TRN2*, like other tetraspanin proteins, are master organizers in so-called 'tetraspanin-enriched microdomains' in the plasmamembrane. So the question is to which extent *TRN2* acts directly and specifically in auxin transport in the context of the paper. The authors show some evidence, however since the *trn2* mutant seems to have a quite pleiotropic phenotype, it is a bit difficult to distinguish direct from indirect (e.g. morphology-related) effects.
3. Figure 4b, c: based on these images it seems that *TRN2* is also activated in the *crc* mutant outside the *CRC* expression domain (?). What is needed here is an in situ hybridization to detect the *CRC* transcript in wildtype next to it.
4. For the key message of the paper, it is important to show that loss of *trn2* function reverses the *crc knu* double mutant phenotype to that of a *knu* single mutant. However, Figure 5 is not very clear in the figures and graphs. Especially, I would like to know in Figure 5J how many plants have a "knu-like" phenotype, instead the authors only differentiate between weak and strong *crc knu*-like phenotypes.
5. Page 10, line 2: the authors say that the *pCRC::TRN2* fruits mimic the *crc* phenotype. However, the p-value of 0.15 suggests that this is not significant.
6. The authors use *TRN2-GFP* reporter gene expression to study subcellular localization of *TRN2*. However it is not clear whether this fusion protein is functional. This should be shown by mutant complementation.
7. Based on the presented data, it is not clear to me how *TRN2* regulates auxin signaling – or auxin transport (while the title of the manuscript suggests a direct, clear relationship)? *PIN1* localization does not seem to be clearly affected based on Figure 2... The results presented in Figure 7 are interesting, but do not distinguish between auxin synthesis and auxin transport.
8. What I find missing are confocal pictures of *PIN1* localization in the *trn2* mutant as well as in double mutant combinations.
9. The p-values throughout the text do not match with the p-values reported in the figures. This is somewhat worrying. What is the difference in p-value calculation? Which one is correct?

Minor comments:

1. page 4, last paragraph, sentence "Double mutants in genetic backgrounds...": add ref. 30
2. Pictures of wildtype plants are missing as reference in all the figures. This makes it difficult to judge specific effects of chemical treatments, and to verify e.g. statements like "pCRC::iaaH fruits exhibited fused carpels and normal style length" – what is meant by "normal"?
3. Figure 4a: mention the time of DEX induction that was used in the legend. Is the y axis in log2 scale?
4. Figure 5 l to n: the floral stages seem to be slightly different

Reviewer #3 (Remarks to the Author):

What are the major claims of the paper?

Yamaguchi et al:

The authors focus on the role of CRC during the termination of the floral meristem and the initiation of the gynoecium from the remnants of the floral meristem. This is an interesting developmental event as it illuminates the mechanism required for proper termination of meristematic fate. AGAMOUS (AG) has previously been shown to repress WUS expression, a key step in the termination process. KNU and CRC are downstream targets of AG regulation. KNU has previously been shown to be required for WUS repression while the role of CRC in this process was incompletely understood. Thus the focus of this paper is on the role of CRC in the termination process.

Here the authors show that KNU and CRC act synergistically to repress WUS expression. The authors claim that CRC functions as part of a mechanistic link that regulates the termination of the floral meristem and the coordinated initiation of the gynoecium. They provide evidence that indicates that CRC functions by fine tuning auxin transport and enabling the generation of proper auxin gradients within the terminating floral meristem. They demonstrate that CRC is able to influence auxin transport as CRC functions as a direct transcriptional repressor of TORNADO2 (TRN2). They propose that TRN2 functions as a membrane localized auxin efflux transporter.

Is the work convincing, and if not, what further evidence would be required to strengthen the conclusions?

1) Role of TRN2 as an auxin efflux transporter: I think that almost all of the conclusions are very well supported. The only exception in my opinion is that the authors claim that auxin transport specifically is being disrupted however it may be that auxin homeostasis more generally is being disrupted. I think that the evidence for TRN2 altering auxin homeostasis is very convincing. The evidence for it as an auxin exporter could be more clearly supported. I think the authors have a choice here. Alter the writing to reduce the emphasis on TRN2 as an auxin transporter or provide additional evidence of that role. I believe that the importance of the paper is not diminished by leaving the language in support of altered auxin homeostasis and leaving the speculation or hypothesis of TRN2 as an exporter to the discussion section. I believe that additional proof such as the direct measurement of auxin efflux in a heterologous system (perhaps in yeast as in Petrusek et al 2006. "PIN Proteins Perform a Rate-Limiting Function in Cellular Auxin Efflux") would still be needed to ascertain with certainty the role of TRN2 as an efflux transporter. Unless these experiments are already carried out and published by another group that I am unaware of, the specific role of TRN2 in efflux has not been established by the data presented here.

In addition to this major point I have additional points where I make suggestions for solidifying the supporting evidence.

2) Figures 2n and 2o are of low quality. It should be possible to get better images by dissecting away the perianth organs and using a confocal. This would make it easier to substantiate the author's claims about expression patterns in these panels. Figures 7g and 7h should be similarly improved.

3) In figure 3 authors claim that local auxin is required for rescue of the *crc knu* phenotype and use CRC promoter to drive the rescue construct. It is still possible that the auxin synthesis in other portions of the *crc knu* mutant would also rescue. Thus the authors should examine the ability of a *pro:PHB::iaaH* construct (or another promoter that drives the *iaaH* outside of the CRC domain) to rescue the phenotype. If this does not rescue, then the claim regarding the location required for auxin replacement can be supported. If a *pro:PHB::iaaH* construct can indeed rescue, then the authors will be able to conclude more generally that auxin synthesis within the gynoecium (regardless of the exact expression domain) is sufficient to rescue the phenotype.

4) Page 8: with respect to Figure 4b and c: the de-repression of TRN2 appears to be throughout the developing gynoecium and even in stamens (i.e. outside the CRC expression domain). Does this suggest a non-cell autonomous effect of CRC on TRN2 (perhaps in addition to the direct effect that the authors argue for based on the ChIP results of Fig 4f.)? Authors should discuss this possibility.

Are they novel and will they be of interest to others in the community and the wider field?

The results are novel and interesting. The paper will be of interest to a wide diversity of plant biologists due to the illumination of the molecular mechanisms of this important developmental event (termination of the floral meristem and the initiation of the female reproductive structure the gynoecium). I also believe the functions of TRN2 have to date been relatively underexplored and this manuscript begins to clarify the role of TRN2 during floral development. The author suggest the possibility of additional YABBY family members functioning in the repression of TRN2 and if this is subsequently shown to be a more general mechanism outside of the flower, this paper would begin to illuminate the roles the YABBY/TRN2 members more generally.

Minor points:

1) It would be nice to see the effects of *crc knu* double on PIN1 expression at stage 5 when the floral meristem is still intact, just before the initiation of the gynoecium.

2) I believe that Larsson et al 2014 (see below) have shown the expression patterns of PIN1 and other PINS during early gynoecium development and should be referenced here on page 6 when introducing Fig 2j. Additionally it would be nice to discuss the author's results in the context of the results from Larsson et al. with respect to the meristematic medial domain of the gynoecium.

Polar auxin transport is essential for medial versus lateral tissue specification and vascular-mediated valve outgrowth in *Arabidopsis* gynoecia. Larsson E, Roberts CJ, Claes AR, Franks RG, Sundberg E. *Plant Physiol.* 2014 Dec;166(4):1998-2012. doi: 10.1104/pp.114.245951

3) Page 5 last paragraph: "363 genes whose expression was altered..." include the phrase "in the *crc-1 knu-1* double mutant relative to the *knu-1* single mutant" for clarity.

4) Fig 1 legend replace "arrowheads" with "asterisk"

5) Authors indicate on page 11 – last line of results – that "results indicate that TRN2 modulates auxin efflux activity. It remains a possibility the TRN2 modulates auxin homeostasis in some other mechanistic fashion such as degradation or conjugation. Authors perhaps should replace indicates with suggests and indicate other possibilities are not ruled out.

6) In first paragraph of the discussion authors use the term network hub genes. I don't think they have evidence for "hub" status (i.e. an investigation of the inputs and outputs of these TFs). They have evidence for the importance of these genes but support for network hubs is not given. Authors might chose to leave "hub" status out of the discussion.

Reviewer #4 (Remarks to the Author):

The manuscript "Fine-tuning of auxin transport governs the transition from floral stem cell maintenance to gynoecium formation" by Yamaguchi et al. (NCOMMS-16-30031) presents their novel findings on the role of transcription factor CRC and its target TRN in regulation of stem cells in floral termination. One of the most exciting findings is the very severe indeterminate *crc knu* double mutant floral phenotype, which was associated with reduced local auxin response and recovered by the auxin transport inhibitor NPA. The authors further identified TRN gene as a biologically relevant target of CRC. In addition, their cell biological studies suggested the function of TRN in modulation of auxin distribution. The presented experimental data consistently support the role of CRC-dependent repression of TRN in regulating floral meristem termination through auxin. These findings are novel and thus extend our knowledge on the mechanism for plant development involving local hormonal regulation. Overall, the research topic is very interesting and the manuscript contains significant findings. However, several minor points need to be improved to meet the high standard of Nature Communications.

Followings are minor comments on the manuscript:

1) Page 8, line 23.

CRC was specifically enriched at the TIII region. Please define 'TIII'.

2) Page 9 line 18.

The number of carpels in each genotype was described. But the graph showing quantified data was not properly cited. Please cite Fig. 5j.

3) Page 10, line 26. Fig. 7b-d.

Localization of TRN2-GFP is not very clear. Images with higher resolution would be preferred. Please specify which tissue in pedicel was observed.

4) Page 10, line 26, 28.

TRN2 should be replaced with TRN2-GFP.

5) P11, line 10.

The authors claimed that TRN2 modulates auxin efflux activity. I tend to agree with this based on the pieces of circumstantial evidence provided by this study. Especially, in *crc* mutant, broad expression of PIN1-GFP was detected supporting increased auxin efflux. But I would suggest to tone down this statement in this particular section, because auxin efflux activity is not directly measured in the experiments with T87 cells and therefore possibility of altered auxin metabolism can not be excluded.

6) Page 13, line 21.

In the 'Methods' section, information on *wus-1* is missing.

7) Page 15, line 15.

Please provide the information about the confocal system (Olympus FV1000?) and objective lens used for imaging.

Point-by-point Responses to referees' comments

Comments to reviewer #1

General comment by Reviewer #1

This manuscript by Yamaguchi et al presents interesting work showing how a target of AG, CRC, is required for floral meristem termination and gynoecium formation through regulation of auxin dynamics. They show how CRC functions with KNU to control determinacy and that CRC directly represses a plasma-membrane protein-encoding gene TRN2 to control auxin dynamics.

The data in the manuscript are solid and provides interesting new knowledge of the transition between meristem maintenance and organ initiation, which will be of interest to a wide audience such as the readers of Nature Communication. In particular, the genetics presented gives unequivocal support to the role of CRC and KNU in repressing WUS expression to promote transition from floral meristem maintenance to gynoecium development. The paper is logically structured, although the writing could be improved. Especially the discussion is not sufficiently concise and will require significant revision. Please see a list of comments below.

General Response

We are grateful to Reviewer #1 for the critical comments, which have helped us improve our paper. As indicated in the following **Responses**, we have taken all of these comments and suggestions into account in the revised version of our manuscript.

Specifically, we improved our discussion by reducing or removing irrelevant parts (e.g., the role of ARF3 and environmental effects) and by adding information about the role of CRC in style formation and floral meristem termination, and the possible roles of CRC downstream target genes other than *TRN2* (see comment from Reviewer 2). Furthermore, we have changed the *DR5* image, included the results of newly performed pharmacological analyses using *crc* and *crc knu*, and performed *pTRN2::GUS* staining in *crc*. We conducted all of the experiments as you suggested. The findings support our conclusions significantly. Please see our point-by-point responses below.

Comment 1 by Reviewer #1

In this work and elsewhere in the literature it is shown that CRC has a role in floral meristem termination as well as in gynoecium patterning and here the authors also convincingly demonstrate that CRC regulates auxin dynamics. However, from the experiments presented in this manuscript it is not clear that CRC controls floral meristem termination via regulation of auxin dynamics. In fact, the lack of DR5 signal at the apex of stage-6 crc gynoecia seems more related to the split style phenotype. Moreover, it is indeed the split style defect that is rescued upon NPA application and in the CRC-iaaH line. It therefore seems that the effect of CRC on auxin is separate from the role of CRC in floral meristem termination with KNU. In my opinion, the authors should make this distinction clearer. This also includes revising the title of the manuscript, which makes the claim that it is the control of auxin transport that governs the floral meristem termination. To my mind that is not demonstrated in the manuscript.

Response

Thank you for pointing out the importance of clarifying floral meristem termination and gynoecium patterning through auxin homeostasis. In the previous version of our paper, we included data showing that the split style defect in *crc* and floral meristem determinacy defect in *crc knu* were both rescued by local auxin production or upon NPA application. To further test whether or not we could distinguish the effect of CRC on floral meristem and gynoecium formation, we treated *crc* and *crc knu* with auxin (IAA, NAA, 2,4-D), auxin signaling inhibitor (PCIB), and another auxin transport inhibitor (1-NOA). These data were included in the revised version of Supplemental Figure 5 and 7. Phenotypic changes were not observed in *crc* and *crc knu* following auxin or PCIB treatment (please also see our response to Comment 3 of Reviewer #3). We described the phenotypic differences between the *crc* mutant subjected to IAAH and auxin treatment. By contrast, the split style defect of *crc* was rescued and the floral meristem determinacy defect of *crc knu* was restored to the *knu* mutant phenotype following 1-NOA treatment.

In the wild-type floral buds at stage 6, *DR5* was expressed in the medial region of the emerging gynoecium, including the tip of the tissues (please also see our response to Minor comment 2 of Reviewer #3). *DR5* expression in that region

was lower in the *crc* mutant background than in the wild type.

Since we could not find out any pharmacological conditions to affect one of phenotypic defects at this point and *DR5* expression in *crc* was affected medial region of arising gynoecium, we would prefer to discuss CRC controls floral meristem termination in *knu* mutant via auxin homeostasis and the effect of CRC on auxin-mediated style formation and floral meristem termination is tightly linked.

Comment 2 by Reviewer #1

*The experiment using *iaaH* is elegant and clearly shows that auxin homeostasis is important in this process. However, on the last line of page 7, the authors specifically claim that these data uncover a role for CRC in auxin transport. This experiment does not show that and it therefore cannot be concluded at this stage whether it is transport, biosynthesis, degradation or signaling that is important.*

Response

We fully agree that the experiment using *iaaH* does not conclude whether auxin transport, metabolism, or signaling is important in this process. Thus, we have rephrased the sentence in the revised version of manuscript.

Comment 3 by Reviewer #1

*The data in Fig. 7i-l are confusing in that the genetics and the model in Fig. 7m suggests that *TRN2* function to inhibit PINs. However, the data in Fig. 7i-l seem to suggest that auxin is transported out of these callus cells and into the medium (how is *PIN1-GFP* localized in these cells?). This apparent contradiction of their model needs to be better explained.*

Response

We apologize for the confusing description of the model diagram (Fig. 7m). We meant to state that *TRN2* inhibits the establishment of auxin maxima—not PINs—in the emerging gynoecium. Because the *DR5* signal was reduced when we overexpressed *TRN2*, we have rephrased the term “auxin” to “auxin maxima” in our model.

In this model, three negative regulations underlie *WUS* repression by AG via auxin. 1) Since CRC directly binds to the *TRN2* promoter and represses its

expression, CRC inhibits *TRN2* expression. 2) Since the *DR5* signal is reduced when *TRN2* is misexpressed, *TRN2* inhibits the proper establishment of auxin maxima. 3) Since auxin maxima and proper repression of *WUS* expression are observed in wild-type plants, but not in *crc* nor *crc knu*, auxin negatively controls *WUS* expression.

Comment 4 by Reviewer #1

The promoter analysis of TRN2 is clear and convincing showing that the YABBY element is important for control of TRN2 expression. Although, TRN2 expression by in situ in WT and crc mutant is presented in Fig. 4b,c, it would be a good to have the TRN2:GUS in crc for comparison to Fig. 4g,h.

Response

In the revised version of our paper, we expressed *pTRN2::GUS* in the *crc* mutant background (Supplemental figure 10h and i). Consistent with the pattern of *TRN2* expression as determined by *in situ* hybridization, *pTRN2::GUS* was misexpressed throughout *crc* carpels.

Comment 5 by Reviewer #1

ARF3 appears prominently in the Discussion although no data on ARF3 are included in the manuscript. However, the arguments for linking ARF3 to AG/CRC/KNU regulation by the authors are sound and based on evidence from the literature. Even so, the reference to similar phenotypes between crc knu and arf3 is curious, since Supplementary Fig. 7 only shows PIN1-GFP and in situs (Supplementary Fig. 13 doesn't seem to exist). It is impossible from these images to make the comparison that the authors do. Also, given the connection between auxin and ARF3, the authors should include a reference to the recent work by Simonini et al. (2016) Genes Dev on the mechanism of ARF3 in auxin signaling.

Response

We agree that we have discussed the link between AG/KNU/CRC and ARF3 too extensively, even though we do not have actual data for ARF3. Thus, we have minimized the discussion related to ARF3 and described the effect of ARF3 on *WUS* transcription only. Also, we have included the citation that describes the connection between auxin and ARF3 (Simonini et al., 2016). Lastly, we

apologize for incorrectly citing Supplementary Fig.13. We have removed this sentence.

Comment 6 by Reviewer #1

Following on from the point above, the writing of the Discussion could be significantly improved. As is, it does not provide a clear summary and discussion of the data. It seems messy and unfounded claims/speculations are made. For example, the suggestion that the robustness of the redundant CRC/KNU pathways may exist to promote adaptation to environmental change is not based on any experiments, although it could easily be tested if crc or knu mutants shows altered responses to e.g. higher temperature. There is also a statement that NPA treatment fully rescues the crc knu double mutant. This is not correct; it merely brings it back to the knu phenotype.

Response

We fully agree that the discussion should focus mainly on the data we have. In the revised version of our text, we have made three major changes to the discussion based on your suggestion. 1) Since the previous version of our manuscript did not discuss the role of CRC in style formation and floral meristem termination, we have included these processes in our interpretation. 2) We have minimized the discussion about the possible link between AG/CRC/KNU and ARF3. 3) We have removed unfounded claims and speculations such as the environmental response and network hub. We have also properly rephrased the sections on the *crc knu* phenotype after NPA treatment or local auxin synthesis by *iaaH* throughout the text.

Minor comment 1 by Reviewer #1

It is mentioned three times on first page of the Introduction that the female reproductive structure is known as the gynoecium. One should be enough.

Response

We have made the requested change.

Minor comment 2 by Reviewer #1

On page 4 in Introduction, auxin transport and PIN proteins are described with a

reference to Okada et al. 1991 (ref. 21). It would be more appropriate to refer to the Gälweiler et al (1998) Science paper where PIN1 is cloned.

Response

We agree that we should have cited the work by Gälweiler et al. In the revised version of our paper, we cited this paper.

Minor comment 3 by Reviewer #1

There are issues with reference numbering. For example, on page 6 ref 39 is used to refer to metabolic pathways producing hormones. This should be ref 40. Also in the second line of the next paragraph. Ref 40 is used to refer to NPA treatment. This should surely be ref 32. In addition, on page 11 ref 32 is used to refer to crc ortholog mutants in other angiosperms, although ref 32 contains no such information. Perhaps the authors meant to refer to Orashakova et al. (2009) Plant J 58: 682-693? There may be other reference mistakes that I haven't spotted.

Response

We have corrected the numbering of references accordingly. We also checked citations throughout the manuscript.

Minor comment 4 by Reviewer #1

On page 6, the authors state they performed an expression analysis of the pPIN1::PIN1-GFP marker line. However, it is not just expression, it is also PIN1-GFP localization.

Response

We have corrected the text accordingly.

Minor comment 4 by Reviewer #1

In Figure 4d, four Brassicaceae species are used in a Phylogenetic shadowing. In Materials and methods C. sativa is listed as Cannabis sativa, which is part of the Cannabaceae family. I assume the authors meant to write Camelina sativa.

Response

We have corrected this error in the methods section.

Minor comment 5 by reviewer #1

There is a reference to a Supplementary Figure 13 on page 12. This figure was not included in the material I received.

Response

We have corrected this mistake. Please also see the response to Comment 5 by Reviewer #1.

Minor comment 6 by reviewer #1

*On page 9, there is a description of the morphological defect of the *trn2-1* mutant with a reference to panels in Fig. 5. However, the reference to Supplementary Fig. 9 (showing qPCR data of *TRN2* expression in WT and *trn2-1*) does not match the text. The expression data should therefore either be mentioned or the figure removed.*

Response

We apologize for not mentioning the *TRN2* mRNA level in the original version of our text. In the revised version of our paper, we have mentioned this result when we presented the *trn2* mutants for the first time.

Comments to reviewer #2

General comment by reviewer #2

Yamaguchi et al. aim to provide a direct molecular link between gynoecium establishment, auxin transport and floral meristem determination. Based on previous research, floral meristem determination linked with WUS repression has been shown to be controlled by several pathways. Until now, only the direct and KNU-mediated repression of WUS by the floral homeotic regulator AG has been reported, while mutant phenotypes suggest additional pathways. Using genetic analyses, the authors show that the known direct AG-target CRC contributes to meristem determinacy and WUS repression additionally to KNU. In gene expression microarray experiments comparing knu-1 single mutants and knu-1 crc-1 mutants, the authors identify 363 significantly different expressed genes. Among those are six genes known to be involved in meristem functions based on GO annotation. For example, CURLY LEAF and SHEPHERD are known to be involved in suppression of WUS activity. Nevertheless, the authors choose to study one gene, TRN2, and its potential role in auxin signaling as candidate pathway by which CRC mediates WUS repression in a non-cell autonomous manner. Treatment of crc-1 and knu-1 mutants with NPA suggest that auxin transport is indeed required for mediating WUS repression by CRC. PIN1 localization however seems to be largely unaffected, besides a mild lateral expansion in the crc mutant. The authors further show that local overproduction of auxin in the CRC expression domain partially rescues the crc knu floral meristem termination defect, and a mutation in trn2 partly suppresses the crc knu meristem termination defect. Finally, the authors provide evidence for a role of TRN2 in auxin signaling.

Overall, the manuscript uses a balanced combination of alternative approaches to elucidate the challenging question on dissecting cellular pathways mediating WUS repression in the floral meristem. However, I have a number of major – especially technical – concerns and questions.

General response

We are grateful to Reviewer #2 for the critical feedback, which has helped us improve our paper. We fully agree that we should address technical problems so that we can provide clear data of high resolution (where applicable). We have

newly included RT-PCR data among the final six CRC candidate genes to show why we focused on *TRN2*, and traditional auxin transport assays in *trn2*, which grew to comparative sizes as the wild type under the same conditions. We repeated the quantification of *knu*-like plants in *trn2 crc knu* to show the importance of *TRN2* in the CRC pathway (and the potential importance of other target(s) downstream of CRC). Also, we have provided the requested controls (CRC mRNA detection by *in situ* hybridization, PIN1-GFP in *crc knu*, and phenotypic rescue of *trn2* by *gTRN2-GFP*) to further support our data. In addition, we have significantly revised the text, especially to highlight the importance of other CRC target candidates, and to tone down our strong statement about the role of *TRN2* on auxin transport. We hope that the revised version of manuscript will now be deemed suitable for publication. Please see our detailed point-by-point responses below.

Comment 1 by Reviewer #2

Page 6: The reasoning by which the authors initially choose to focus only on TRN2 as potentially mediating the role of CRC in WUS repression is not well supported in my point of view. For example, SHEPHERD plays a role in spatial restriction of WUS expression (via the CLV1 pathway). Among the 363 differentially expressed genes, there might well be more (yet uncharacterized) genes that are potential candidates directly downstream of CRC.

Response

We fully agree that we should have provided a better reasoning as to why we focused on *TRN2*. We actually pre-screened the final six CRC downstream target genes by qRT-PCR. Based on qRT-PCR, *TRN2* was greatly upregulated in *crc knu* compared to the *knu* single mutant. In the revised version of our paper, we have included these data in Supplemental figure 9.

Since we observed only partial rescue of the *trn2 crc knu* triple mutant, it is obvious that *TRN2* is not the sole downstream target of CRC to control floral meristem termination. As you pointed out, it is entirely possible that some other candidate genes (including the final six CRC candidate target genes) identified by expression profiling act downstream of CRC and in parallel with *TRN2*. In the revised version of our discussion, we have discussed this possibility. Moreover,

we added factor X and Y in our pathway model to show this possibility more clearly. Further introduction of mutations of the CRC downstream target genes into the *trn2 crc knu* triple mutant would clarify the importance of other CRC targets for floral meristem determinacy.

Comment 2 by Reviewer #2

*The role of TRN2 in auxin transport/signaling is far from understood. The pleiotropic mutant phenotype suggests as well auxin-independent roles, and to my knowledge the current idea is that TRN2, like other tetraspanin proteins, are master organizers in so-called 'tetraspanin-enriched microdomains' in the plasmamembrane. So the question is to which extent TRN2 acts directly and specifically in auxin transport in the context of the paper. The authors show some evidence, however since the *trn2* mutant seems to have a quite pleiotropic phenotype, it is a bit difficult to distinguish direct from indirect (e.g. morphology-related) effects.*

Response

The *trn2* mutant shows an auxin-related phenotype as well as a twisted phenotype, which is not observed in typical auxin-deficient mutants. Thus, we fully agree that TRN2 may have general roles other than auxin homeostasis. To demonstrate the role of TRN2 in auxin transport, we newly performed an auxin transport assay using wild-type and *trn2* plants. To minimize the effects of differences in tissue composition between the wild type and *trn2* in the auxin transport assay, the wild type and *trn2* mutant were grown on MS medium. Under this growth condition, morphological differences between the wild type and *trn2* mutant were much smaller than those in plants grown on soil by unknown mechanism (Supplemental figure 13). Even by using these plant materials, reduced auxin transport capacity was observed in *trn2* mutant compared to the wild type. Although we have tried to minimize the effects of tissue composition between the wild type and *trn2* in the auxin transport assay, our results could still be linked to this issue. Furthermore, the possibility that auxin homeostasis is altered in this mutant can not be excluded. Thus, we have toned down our strong statement and stated that TRN2 affects auxin homeostasis in the revised version of our manuscript.

Comment 3 by Reviewer #2

Figure 4b, c: based on these images it seems that TRN2 is also activated in the crc mutant outside the CRC expression domain (?). What is needed here is an in situ hybridization to detect the CRC transcript in wildtype next to it.

Response

We have included the results of a *CRC* mRNA *in situ* hybridization at the same stage of floral buds shown in Fig. 4b and c in Supplemental data 9g. Also, we have included a discussion explaining why *TRN2* is activated in the *crc* mutant outside the *CRC* expression domain in the revised version of our result and discussion. (Please see Comment 4 by Reviewer #3).

Comment 4 by Reviewer #2

For the key message of the paper, it is important to show that loss of trn2 function reverses the crc knu double mutant phenotype to that of a knu single mutant. However, Figure 5 is not very clear in the figures and graphs. Especially, I would like to know in Figure 5J how many plants have a “knu-like” phenotype, instead the authors only differentiate between weak and strong crc knu-like phenotypes.

Response

Introducing the *trn2* mutation into *crc knu* partially rescues the mutant phenotypes, but we did not identify the triple mutant flowers showing *knu*-like phenotypes, as long as we observed the first five fruits from six individual plants in the original version of our manuscript. Therefore, we increased the number of plants and observed the first five fruits from twenty individual plants in the revised version of our paper. The *knu*-like phenotype was observed in the *trn2 crc knu* triple mutant at a frequency of approximately 3.0%, while such a phenotype was never observed in *crc knu* double mutants. We have replaced old data with a new image showing this in Fig. 5j.

Since we saw partial rescue in the *trn2 crc knu* triple mutant compared to *crc knu*, multiple targets downstream of *CRC* are most likely to be involved in floral meristem termination (as you point out in Comment 1). Thus, we have added this quantitative description of the triple mutants and have also included discussion to emphasize the involvement of *CRC* targets other than *TRN2*. Finally, we

added factor X and Y to our pathway model to clearly indicate the importance of other CRC targets.

Comment 5 by Reviewer #2

Page 10, line 2: the authors say that the pCRC::TRN2 fruits mimic the crc phenotype. However, the p-value of 0.15 suggests that this is not significant.

Response

We apologize for our confusing description of the *p*-value for the phenotype of *crc* and *pCRC::TRN2* in the original version. We meant to say that there was no significant difference between *pCRC::TRN2* and the *crc* mutant. In the revised version of our paper, we have clarified the point that there is no phenotypic difference between *crc* and *pCRC::TRN2* (*p* = 0.15). This result suggests that misexpression of *TRN2* under the control of the *CRC* promoter can mimic *crc* phenotypes. Furthermore, we have included wild-type data, showing significant differences between the wild type and *pCRC::TRN2* based on a Chi-square test in Fig. 6e (*p* = 4.3×10^{-13}).

Comment 6 by Reviewer #2

The authors use TRN2-GFP reporter gene expression to study subcellular localization of TRN2. However, it is not clear whether this fusion protein is functional. This should be shown by mutant complementation.

Response

We agree that we should have shown mutant complementation to confirm that our TRN2-GFP fusion protein was functional. In the revised version of our paper, we have included the *trn2* rescue experiment by introducing TRN2-GFP (Supplementary Fig. 13a-f). The TRN2-GFP transgene indeed fully rescued the small and twisted phenotype of *trn2*.

Comment 7 by Reviewer #2

Based on the presented data, it is not clear to me how TRN2 regulates auxin signaling – or auxin transport (while the title of the manuscript suggests a direct, clear relationship)? PIN1 localization does not seem to be clearly affected based on Figure 2... The results presented in Figure 7 are interesting, but do not

distinguish between auxin synthesis and auxin transport.

Response

We fully agree that it is difficult to conclude that TRN2 controls auxin transport, metabolism, or signaling. To confirm the role of TRN2 in auxin transport, we have performed the auxin transport assay using wild-type and *trn2* mutant plants. The *trn2* mutant had a reduced auxin transport capacity compared to the wild type. However, we cannot exclude the possibility that auxin metabolism or signaling are altered in this mutant. Thus, we have toned down our strong statement (as Reviewers 3 and 4 also suggested) and stated that TRN2 controls auxin maxima in the revised version of our manuscript.

Comment 8 by Reviewer #2

*What I find missing are confocal pictures of PIN1 localization in the *trn2* mutant as well as in double mutant combinations.*

Response

1) In the original version of our paper, we showed the PIN1-GFP data in the *trn2* mutant in Supplemental figure 11. To highlight it, we moved these data to Figure 7g and h. Further, we have included data based on the expression of PIN1-GFP in *crc knu* in Supplemental figure 8e–h. We observed that the PIN1 expression domain in *trn2* was smaller than in wild-type plants. This result is consistent with our other data showing that the *crc* mutant, which has more *TRN2* transcripts, has an expanded PIN1 expression domain. Also, signals of PIN1-GFP cellular localization in the *trn2* background were weaker than in the wild type. We have explained these data in the revised version of our manuscript. PIN1 expression and localization in *crc knu* were similar to those in *crc*. In accord with the elevated expression of *TRN2*, broad expression of PIN1-GFP was detected.

Comment 9 by Reviewer #2

The p-values throughout the text do not match with the p-values reported in the figures. This is somewhat worrying. What is the difference in p-value calculation? Which one is correct?

Response

We apologize for the confusion. In the original version of our figures, we provided

the exact p -values. On the other hand, the p -values in the original version of text were shown by using inequality signs. Thus, even though the number itself shown in the figures and text is different, the results were essentially the same. To avoid this confusion, we have described the exact p -values in the revised version of the text.

Minor comment 1 by Reviewer #2

page 4, last paragraph, sentence “Double mutants in genetic backgrounds...”: add ref. 30

Response

We have modified the text as you suggested.

Minor comment 2 by Reviewer #2

Pictures of wildtype plants are missing as reference in all the figures. This makes it difficult to judge specific effects of chemical treatments, and to verify e.g. statements like “pCRC::iaaH fruits exhibited fused carpels and normal style length” – what is meant by “normal”?

Response

We fully agree that the previous version of our paper was difficult to understand. We have included a brief explanation of the wild-type, *crc*, *knu*, and *crc knu* phenotype before presenting our detailed analyses. In addition, we have changed a couple of sentences to specify the controls clearly.

Minor comment 3 by Reviewer #2

Figure 4a: mention the time of DEX induction that was used in the legend. Is the y axis in log2 scale?

Response

A linear scale was used for the y-axis in the original version of Fig. 4a. We agree that we should show statistical difference and data distribution. In the revised version of our paper, we show expression data by box plot. We performed three independent *TRN2* expression tests using qRT-PCR in *crc* and CRC-GR after a single treatment with 10 μ M DEX and included these data in the revised version of our paper. We observed significant differences, using Student's *t*-tests, based

on three independent RNA samples and three technical replicates for each RNA sample (i.e., 3 x 3 samples). The p -value was 1.0×10^{-4} .

Minor comment 4 by Reviewer #2

Figure 5 l to n: the floral stages seem to be slightly different.

Response

We agree that the *crc knu* floral buds shown in Fig. 5m appears to be at stage 6, while *trn2-1* and *trn2-1 crc-1 knu-1* flowers are at stage 7. In the revised version of our manuscript, we have included an *in situ* hybridization image showing the localization of *WUS* in a stage 7 flower of *crc-1 knu-1* to unify the floral stages of the three genotypes.

Comments to Reviewer #3

General comment by Reviewer #3

The authors focus on the role of CRC during the termination of the floral meristem and the initiation of the gynoecium from the remnants of the floral meristem. This is an interesting developmental event as it illuminates the mechanism required for proper termination of meristematic fate. AGAMOUS (AG) has previously been shown to repress WUS expression, a key step in the termination process. KNU and CRC are downstream targets of AG regulation. KNU has previously been shown to be required for WUS repression while the role of CRC in this process was incompletely understood. Thus the focus of this paper is on the role of CRC in the termination process.

Here the authors show that KNU and CRC act synergistically to repress WUS expression. The authors claim that CRC functions as part of a mechanistic link that regulates the termination of the floral meristem and the coordinated initiation of the gynoecium. They provide evidence that indicates that CRC functions by fine tuning auxin transport and enabling the generation of proper auxin gradients within the terminating floral meristem. They demonstrate that CRC is able to influence auxin transport as CRC functions as a direct transcriptional repressor of TORNADO2 (TRN2). They propose that TRN2 functions as a membrane localized auxin efflux transporter. The results are novel and interesting.

The paper will be of interest to a wide diversity of plant biologists due to the illumination of the molecular mechanisms of this important developmental event (termination of the floral meristem and the initiation of the female reproductive structure the gynoecium). I also believe the functions of TRN2 have to date been relatively underexplored and this manuscript begins to clarify the role of TRN2 during floral development. The author suggest the possibility of additional YABBY family members functioning in the repression of TRN2 and if this is subsequently shown to be a more general mechanism outside of the flower, this paper would begin to illuminate the roles the YABBY/TRN2 members more generally.

General response

We are grateful to Reviewer #3 for the constructive feedback, which has helped us improve our paper. As indicated in the following responses, we have taken all of these comments and suggestions into account in the revised version of our

manuscript. Please see our point-by-point responses below.

Comment 1 by Reviewer #3

Role of TRN2 as an auxin efflux transporter: I think that almost all of the conclusions are very well supported. The only exception in my opinion is that the authors claim that auxin transport specifically is being disrupted however it may be that auxin homeostasis more generally is being disrupted. I think that the evidence for TRN2 altering auxin homeostasis is very convincing. The evidence for it as an auxin exporter could be more clearly supported. I think the authors have a choice here. Alter the writing to reduce the emphasis on TRN2 as an auxin transporter or provide additional evidence of that role. I believe that the importance of the paper is not diminished by leaving the language in support of altered auxin homeostasis and leaving the speculation or hypothesis of TRN2 as an exporter to the discussion section. I believe that additional proof such as the direct measurement of auxin efflux in a heterologous system (perhaps in yeast as in Petrasek et al 2006. "PIN Proteins Perform a Rate-Limiting Function in Cellular Auxin Efflux") would still be needed to ascertain with certainty the role of TRN2 as an efflux transporter. Unless these experiments are already carried out and published by another group that I am unaware of, the specific role of TRN2 in efflux has not been established by the data presented here.

Response

We fully agree that our data are not sufficient to claim that TRN2 is an auxin efflux transporter. As you suggested, a heterologous system would be the best way to evaluate the role of TRN2 as an auxin efflux transporter. However, it was technically difficult for us to perform such an analysis. Instead of conducting a heterologous assay, we have performed a traditional auxin transport assay using the wild type and *trn2* mutants and observed that auxin transport is reduced in *trn2* compared to the wild type, as expected. However, we cannot exclude the possibility that this reduction is due to differences in auxin metabolism in the *trn2* mutant. Thus, we have toned down our strong statement, as you and Reviewer 4 suggested, and have stated that TRN2 controls auxin maxima in the revised version of our manuscript.

Comment 2 by Reviewer #3

Figures 2n and 2o are of low quality. It should be possible to get better images by dissecting away the perianth organs and using a confocal. This would make it easier to substantiate the author's claims about expression patterns in these panels. Figures 7g and 7h should be similarly improved.

Response

In the revised version our paper, we have included new *DR5* data derived from experiments in the *crc* and *pCRC::TRN2* background (Insets of figure 2 and 7). Consistent with the previous results, *DR5* expression was reduced in *crc* and *pCRC::TRN2*.

Comment 3 by Reviewer #3

*In figure 3 authors claim that local auxin is required for rescue of the *crc knu* phenotype and use *CRC* promoter to drive the rescue construct. It is still possible that the auxin synthesis in other portions of the *crc knu* mutant would also rescue. Thus the authors should examine the ability of a *pro:PHB::iaaH* construct (or another promoter that drives the *iaaH* outside of the *CRC* domain) to rescue the phenotype. If this does not rescue, then the claim regarding the location required for auxin replacement can be supported. If a *pro:PHB::iaaH* construct can indeed rescue, then the authors will be able to conclude more generally that auxin synthesis within the gynoecium (regardless of the exact expression domain) is sufficient to rescue the phenotype.*

Response

To test whether a general increase in auxin levels within the gynoecium is sufficient to rescue the *crc* and *crc knu* phenotypes, we treated the *crc* and *crc knu* developing flowers with auxin. Although we used three different versions of auxin (IAA, NAA, or 2,4-D), no phenotypic rescues were observed in *crc* or *crc knu* (Supplemental figure 5 and 7). At the same time, we found that the auxin transport inhibitors NPA and 1-NOA could rescue *crc* and *crc knu*. Together with the results of local auxin synthesis by *iaaH*, we favor the claim that auxin replacement or establishment of auxin maxima in the gynoecium are required for phenotypic rescue.

Comment 4 by Reviewer #3

Page 8: with respect to Figure 4 b and c: the de-repression of *TRN2* appears to be throughout the developing gynoecium and even in stamens (i.e. outside the *CRC* expression domain). Does this suggest a non-cell autonomous effect of *CRC* on *TRN2* (perhaps in addition to the direct effect that the authors argue for based on the ChIP results of Fig 4f.)? Authors should discuss this possibility.

Response

Since the original version of Figure 4b and c appeared to be overstained, we have replaced these images. Regardless of the three different assays used to evaluate the wild type and *crc* mutant (in situ, GUS, and GFP), we repeatedly observed that *TRN2* was de-repressed throughout the developing gynoecium. We mentioned a few hypotheses to explain why *TRN2* is activated in the *crc* mutant outside of the *CRC* expression domain. One hypothesis is that the mobile signal is derived from *CRC* activity, since all tested YABBY genes have non-cell-autonomous activities. Auxin could be one such signal. Alternatively, feedback regulation of *TRN2* through auxin could occur after de-repression of *TRN2* at the abaxial regions of developing carpel primordia. Otherwise, other *TRN2* activators such as *ERF115* and *PRR5*, and *CRC* may act antagonistically to control *TRN2* expression.

Minor comment 1 by Reviewer #3

It would be nice to see the effects of *crc knu* double on *PIN1* expression at stage 5 when the floral meristem is still intact, just before the initiation of the gynoecium.

Response

We have included *PIN1* expression and localization data in stage 5 wild-type and *crc knu* flowers in the revised version of Supplemental figure 8e and f. We did not see an obvious difference between the mutant and wild type in *PIN1*-GFP expression and localization at stage 5. At later stages, we observed a difference in *PIN1*-GFP expression. This is reasonable, since it has been reported that *CRC* mRNA is expressed from stage 6 onward (Bowman and Smyth, 1999 Development, 126, 2387-2396).

Minor comment 2 by Reviewer #3

I believe that Larsson et al 2014 (see below) have shown the expression patterns of PIN1 and other PINs during early gynoecium development and should be referenced here on page 6 when introducing Fig 2j. Additionally it would be nice to discuss the author's results in the context of the results from Larsson et al. with respect to the meristematic medial domain of the gynoecium. Polar auxin transport is essential for medial versus lateral tissue specification and vascular-mediated valve outgrowth in Arabidopsis gynoecia. Larsson E, Roberts CJ, Claes AR, Franks RG, Sundberg E. Plant Physiol. 2014 Dec;166(4):1998-2012. doi: 10.1104/pp.114.245951

Response

Thank you for pointing out this important reference. We have cited Larsson et al., 2014 prior to describing our PIN1 expression analysis and also in the discussion. We have also included discussion about the meristematic medial domain of the gynoecium.

Minor comment 3 by Reviewer #3

Page 5 last paragraph: "363 genes whose expression was altered..." include the phrase "in the crc-1 knu-1 double mutant relative to the knu-1 single mutant" for clarity.

Response

We have corrected this sentence.

Minor comment 4 by Reviewer #3

Fig 1 legend replace "arrowheads" with "asterisk"

Response

We have corrected this sentence.

Minor comment 5 by Reviewer #3

Authors indicate on page 11 – last line of results – that "results indicate that TRN2 modulates auxin efflux activity. It remains a possibility the TRN2 modulates auxin homeostasis in some other mechanistic fashion such as degradation or conjugation. Authors perhaps should replace indicates with suggests and

indicate other possibilities are not ruled out.

Response

We fully agree that our data are not sufficient to claim that TRN2 is an auxin efflux transporter. Although we performed traditional auxin transport assays using wild-type and *trn2* mutants and observed reduced auxin transport capacity in the *trn2* mutant, it is possible that TRN2 modulates auxin metabolism. Since we observed changes in auxin maxima in TRN2-misexpressed culture cells or intact plants, we stated that TRN2 controls auxin maxima by regulating auxin homeostasis in the revised version of our manuscript.

Minor comment 6 by Reviewer #3

In first paragraph of the discussion authors use the term network hub genes. I don't think they have evidence for "hub" status (i.e. an investigation of the inputs and outputs of these TFs). They have evidence for the importance of these genes but support for network hubs is not given. Authors might chose to leave "hub" status out of the discussion.

Response

We have excluded any mention of "hub" status in the discussion.

Comments to reviewer #4

General comment by reviewer #4

The manuscript “Fine-tuning of auxin transport governs the transition from floral stem cell maintenance to gynoecium formation” by Yamaguchi et al. (NCOMMS-16-30031) presents their novel findings on the role of transcription factor CRC and its target TRN in regulation of stem cells in floral termination. One of the most exciting findings is the very severe indeterminate crc knu double mutant floral phenotype, which was associated with reduced local auxin response and recovered by the auxin transport inhibitor NPA. The authors further identified TRN gene as a biologically relevant target of CRC. In addition, their cell biological studies suggested the function of TRN in modulation of auxin distribution. The presented experimental data consistently support the role of CRC-dependent repression of TRN in regulating floral meristem termination through auxin. These findings are novel and thus extend our knowledge on the mechanism for plant development involving local hormonal regulation. Overall, the research topic is very interesting and the manuscript contains significant findings. However, several minor points need to be improved to meet the high standard of Nature Communications.

General response

We thank Reviewer #4 for the constructive feedback, which has helped us improve the quality of our paper. As indicated in the following responses, we have taken all of these comments and suggestions into account in the revised version of our manuscript. Please see our point-by-point responses below.

Minor comment 1 by reviewer #4

Page 8, line 23. CRC was specifically enriched at the TIII region. Please define ‘TIII’.

Response

TIII refers to the PCR amplicon used for ChIP and are shown in the figure 4e. We have included an explanation in the figure legend to figure 4e.

Minor comment 2 by reviewer #4

Page 9 line 18. The number of carpels in each genotype was described. But the

graph showing quantified data was not properly cited. Please cite Fig. 5j.

Response

We have properly cited the quantified data in the revised version of our paper.

Minor comment 3 by reviewer #4

Page 10, line 26. Fig. 7b-d. Localization of TRN2-GFP is not very clear. Images with higher resolution would be preferred. Please specify which tissue in pedicel was observed.

Response

In the revised version, we have included better TRN2-GFP images in Supplemental figure 13 g and h. We apologize for our confusing description. We meant to state that the lower part of the flower primordium at stage 2 was imaged. We have corrected this in the revised version of our manuscript.

Minor comment 4 by reviewer #4

Page 10, line 26, 28. TRN2 should be replaced with TRN2-GFP.

Response

We have corrected this error.

Minor comment 5 by reviewer #4

*P11, line 10. The authors claimed that TRN2 modulates auxin efflux activity. I tend to agree with this based on the pieces of circumstantial evidence provided by this study. Especially, in *crc* mutant, broad expression of PIN1-GFP was detected supporting increased auxin efflux. But I would suggest to tone down this statement in this particular section, because auxin efflux activity is not directly measured in the experiments with T87 cells and therefore possibility of altered auxin metabolism can not be excluded.*

Response

We fully agree that our data are not sufficient to claim that TRN2 is an auxin efflux transporter. Although we performed a traditional auxin transport assay in both the wild type and *trn2* mutants and observed reduced auxin transport capacity in the *trn2* mutant, we can not exclude the possibility that auxin metabolism is altered in this mutant. Thus, we have toned down the text and stated that TRN2 controls

auxin maxima by regulating auxin homeostasis in the revised version of our manuscript.

Minor comment 6 by reviewer #4

Page 13, line 21. In the 'Methods' section, information on *wus-1* is missing.

Response

We have added information on *wus-1* in the revised version of our manuscript.

Minor comment 7 by reviewer #4

Page 15, line 15. Please provide the information about the confocal system (Olympus FV1000?) and objective lens used for imaging.

Response

In the revised version of our methods, we have included the name of the confocal microscope (Olympus FV1000) and objective lens (Olympus UPlanSApo) used.

REVIEWERS' COMMENTS:

Reviewer #1 (Remarks to the Author):

Yamaguchi et al have submitted a revised version of their manuscript on the role of auxin dynamics in controlling the transition from floral meristem to gynoecium formation. They have done a thorough job in addressing all the issues that I and the other reviewers have raised to their first version and this has resulted in a much improved and exciting manuscript. I have no further issues to raise.

Reviewer #2 (Remarks to the Author):

The revised version of Yamaguchi et al has significantly improved. I have no more comments.

Reviewer #3 (Remarks to the Author):

The authors have addressed my major and minor concerns to my satisfaction. They have changed the emphasis from auxin transport to auxin homeostasis as was suggested by several reviewers. Other suggestions have also been incorporated.

Reviewer #4 (Remarks to the Author):

I am satisfied with the revised version. I now recommend it for publication in Nature Communications .